# Theoretical characterisation of the Gauss-Newton conditioning in Neural Networks

**Jim Zhao***
University of Basel, Switzerland
jim.zhao@unibas.ch

**Sidak Pal Singh***
ETH Zürich, Switzerland
sidak.singh@inf.ethz.ch

**Aurelien Lucchi**
University of Basel, Switzerland
aurelien.lucchi@unibas.com

## Abstract

The Gauss-Newton (GN) matrix plays an important role in machine learning, most evident in its use as a preconditioning matrix for a wide family of popular adaptive methods to speed up optimization. Besides, it can also provide key insights into the optimization landscape of neural networks. In the context of deep neural networks, understanding the GN matrix involves studying the interaction between different weight matrices as well as the dependencies introduced by the data, thus rendering its analysis challenging. In this work, we take a first step towards theoretically characterizing the conditioning of the GN matrix in neural networks. We establish tight bounds on the condition number of the GN in deep linear networks of arbitrary depth and width, which we also extend to two-layer ReLU networks. We expand the analysis to further architectural components, such as residual connections and convolutional layers. Finally, we empirically validate the bounds and uncover valuable insights into the influence of the analyzed architectural components.

## 1 Introduction

The curvature is a key geometric property of the loss landscape, which is characterized by the Hessian matrix or approximations such as the Gauss-Newton (GN) matrix, and strongly influences the convergence of gradient-based optimization methods. In the realm of deep learning, where models often have millions of parameters, understanding the geometry of the optimization landscape is essential to understanding the effectiveness of training algorithms. The Hessian matrix helps identify the directions in which the loss function changes most rapidly, aiding in the selection of appropriate learning rates and guiding optimization algorithms to navigate the complex, high-dimensional space of parameters. However, in practice, the Hessian is not easily accessible due to the high computational cost and memory requirements. Instead, the GN matrix (or its diagonal form) is commonly employed in adaptive optimization methods such as Adam [Kingma and Ba, 2014] with the goal to improve the conditioning of the landscape. Although the Gauss-Newton matrix $\mathbf{G}_O$ is only an approximation to the full Hessian matrix, it does seem to capture the curvature of the loss very well given the success of many second-order optimization methods based on approximations of the Gauss-Newton matrix, such as K-FAC [Martens and Grosse, 2020], Shampoo [Gupta et al., 2018] or Sophia [Liu et al., 2023]. Particularly interesting is the last method, in which the authors observe that their optimizer based on the Gauss-Newton matrix performs even better than their optimizer based on the full Hessian matrix, implying that the Gauss-Newton matrix is a good preconditioner and captures the curvature of the loss landscape well.

---

*First two authors have equal contribution

38th Conference on Neural Information Processing Systems (NeurIPS 2024).

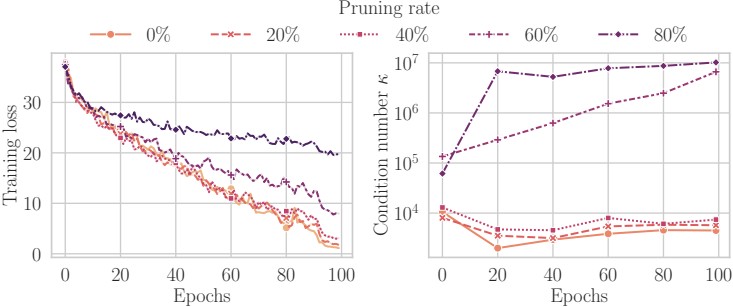

Figure 1: Training loss (left) and condition number $\kappa$ of GN (right) for a ResNet20 trained on a subset of Cifar10 ($n = 1000$) with different proportions of pruned weights. Weights were pruned layerwise by magnitude at initialization.

The prevalence of adaptive optimizers and their success in training neural networks illustrates the ability of the GN matrix to capture curvature information similar to that of the Hessian matrix.

Although the non-convexity of neural loss landscapes can be considered as a given, some landscapes can be tougher to traverse than others. This can, for instance, arise due to, or be amplified by, aspects such as: (i) disparate scales of the input data and intermediate features, (ii) initializing the optimization process at a degenerate location in the landscape (imagine an entire set of neurons being dead), (iii) architectural aspects such as width, depth, normalization layers, etc.

Given the close relation of the GN matrix to the network Jacobian (with respect to the parameters) and hence signal propagation [Lou et al., 2022], its significance also reaches more broadly to problems such as model pruning and compression. For instance, it has been extensively observed empirically that training extremely sparse neural networks from scratch poses a significantly greater challenge compared to pruning neural networks post-training [Evci et al., 2019]. In an illustrative example shown in Fig. 1, we observe that training sparse networks from scratch using stochastic gradient descent results in a slowdown of training, which is also reflected in an increase in the condition number of the GN matrix (similar experiments on Vision Transformers and other architectures can be found in Appendix I.1). This underscores the relevance of the conditioning of the GN matrix for understanding the behaviour of gradient-based optimizers and emphasizes the significance of maintaining a well-behaved loss landscape of the neural network throughout training.

In fact, many fundamental components of deep learning frameworks, such as skip-connections as well as various normalization techniques [Ioffe and Szegedy, 2015, Salimans and Kingma, 2016, Ba et al., 2016, Wu and He, 2018, Miyato et al., 2018], have been to some degree designed to mitigate the challenges posed by ill-conditioning in neural network optimization. This line of work continues to expand to this date, seeking out better normalization schemes, optimization maneuvers, and regularization strategies [Kang et al., 2016, Wan et al., 2013] that allow for easier and faster optimization while avoiding directions of pathological curvature. But despite the numerous approaches to redress the poor conditioning of the landscape, understanding the precise factors of ill-conditioning within the network structure, and their relative contributions, has remained largely underdeveloped.

Hence, our aim in this work is to carry out a detailed theoretical analysis of how the conditioning of the Gauss-Newton matrix is shaped by constituting structural elements of the network — i.e., the hidden-layer width, the depth, and the presence of skip connections, to name a few. We will shortly formally introduce the Gauss-Newton matrix and how it is connected to the Hessian of the loss function. Concretely, we would like to provide tight bounds for the condition number of these two terms as a function of the spectra of the various weight matrices and reveal the interplay of the underlying architectural parameters on conditioning. Furthermore, we aim to investigate the impact of both the dataset's structure and the initialization procedure on the conditioning of the loss landscape.

**Contributions.** Taking inspiration from prior theoretical analyses of deep linear networks, we make a first foray into this problem by rigorously investigating and characterizing the condition number of the GN matrix for linear neural networks (the extension to the second term of the Gauss-Newton decomposition is touched upon in the Appendix). Our analysis holds for arbitrary-sized networks and unveils the intriguing interaction of the GN matrix with the conditioning of various sets of individual layers and the data-covariance matrix. We also complement our analysis in the linear case by studying the effect of non-linearities, via the Leaky-ReLU activation, albeit for two-layer networks. Importantly, as a consequence of our analysis, we show the precise manner in which residual networks

with their skip connections or batch normalization can help enable better conditioning of the loss landscape. While our work builds on Singh et al. [2021], our main contribution is the introduction of tight upper bounds for the condition number of the Gauss-Newton (GN) matrix for linear and residual networks of arbitrary depth and width. To the best of our knowledge, this has not been addressed in the literature before. Lastly, given that our bounds are agnostic to the specific values of the parameters in the landscape, we show the phenomenology of conditioning during the training procedure and the corresponding validity of our bounds.

## 2 Setup and background

**Setting.** Suppose we are given an i.i.d. dataset $S = \{(\mathbf{x}_1, \mathbf{y}_1), \ldots, (\mathbf{x}_n, \mathbf{y}_n)\}$, of size $|S| = n$, drawn from an unknown distribution $p_{\mathbb{X}, \mathbb{Y}}$, consisting of inputs $\mathbf{x} \in \mathbb{X} \subseteq \mathbb{R}^d$ and targets $\mathbf{y} \in \mathbb{Y} \subseteq \mathbb{R}^k$. Based on this dataset $S$, consider we use a neural network to learn the mapping from the inputs to the targets, $F_{\boldsymbol{\theta}} : \mathbb{X} \mapsto \mathbb{Y}$, parameterized by $\boldsymbol{\theta} \in \boldsymbol{\Theta} \subseteq \mathbb{R}^p$. To this end, we follow the framework of Empirical Risk Minimization [Vapnik, 1991], and optimize a suitable loss function $L : \boldsymbol{\Theta} \mapsto \mathbb{R}$. In other words, we solve the following optimization problem,

$$\boldsymbol{\theta}^{\star} = \operatorname*{argmin}_{\boldsymbol{\theta} \in \boldsymbol{\Theta}} \; L(\boldsymbol{\theta}) = \frac{1}{n} \sum_{i=1}^{n} \ell\left(\boldsymbol{\theta}; (\mathbf{x}_i, \mathbf{y}_i)\right) ,$$

say with a first-order method such as (stochastic) gradient descent and the choices for $\ell$ could be the mean-squared error (MSE), cross-entropy (CE), etc. For simplicity, we will stick to the MSE loss.

**Gauss-Newton Matrix.** We analyze the properties of the outer gradient product of the loss function, $\mathbf{G}_{\mathrm{O}}$, which we call the Gauss-Newton matrix, defined as

$$\mathbf{G}_{\mathrm{O}} = \frac{1}{n} \sum_{i=1}^{n} \nabla_{\boldsymbol{\theta}} F_{\boldsymbol{\theta}}(\mathbf{x}_i) \, \nabla_{\boldsymbol{\theta}} F_{\boldsymbol{\theta}}(\mathbf{x}_i)^{\top}, \tag{1}$$

where, $\nabla_{\boldsymbol{\theta}} F_{\boldsymbol{\theta}}(\mathbf{x}_i) \in \mathbb{R}^{p \times k}$ is the Jacobian of the function with respect to the parameters $\boldsymbol{\theta}$, $p$ is the number of parameters, $k$ is the number of outputs or targets. This outer product of the gradient is closely related to the Hessian of the loss function via the Gauss-Newton decomposition [Schraudolph, 2002, Sagun et al., 2017, Martens, 2020, Botev, 2020], hence the chosen name, which decomposes the Hessian via the chain rule as a sum of the following two matrices:

$$\mathbf{H}_{\mathrm{L}} = \mathbf{H}_{\mathrm{O}} + \mathbf{H}_{\mathrm{F}} = \frac{1}{n} \sum_{i=1}^{n} \nabla_{\boldsymbol{\theta}} F_{\boldsymbol{\theta}}(\mathbf{x}_i) \left[ \nabla_{F_{\boldsymbol{\theta}}}^2 \ell_i \right] \nabla_{\boldsymbol{\theta}} F_{\boldsymbol{\theta}}(\mathbf{x}_i)^{\top} + \frac{1}{n} \sum_{i=1}^{n} \sum_{c=1}^{K} [\nabla_{F_{\boldsymbol{\theta}}} \ell_i]_c \, \nabla_{\boldsymbol{\theta}}^2 F_{\boldsymbol{\theta}}^c(\mathbf{x}_i),$$

where $\nabla_{F_{\boldsymbol{\theta}}}^2 \ell_i \in \mathbb{R}^{k \times k}$ is the Hessian of the loss with respect to the network function, at the $i$-th sample. Note that if $\ell_i$ is the MSE loss, then $\mathbf{H}_{\mathrm{O}} = \mathbf{G}_{\mathrm{O}}$.

**Remark R1** (Difference between $\mathbf{H}_L$ and $\mathbf{G}_O$). *When considering MSE loss, the difference between the Gauss Newton matrix $\mathbf{G}_O$ and the Hessian of the loss function $\mathbf{H}_L$ depends on both the residual and the curvature of the network $F_{\boldsymbol{\theta}}(\mathbf{x})$. Thus, close to convergence when the residual becomes small, the contribution of $\mathbf{H}_F$ will also be negligible and $\mathbf{G}_O$ is essentially equal to $\mathbf{H}_L$. Furthermore, Lee et al. [2019] show that sufficiently wide neural networks of arbitrary depth behave like linear models during training with gradient descent. This implies that the Gauss-Newton matrix is a close approximation of the full Hessian in this regime throughout training.*

**Condition number and its role in classical optimization.** Consider we are given a quadratic problem, $\operatorname{argmin}_{\mathbf{w} \in \mathbb{R}^p} \frac{1}{2} \mathbf{w}^{\top} \mathbf{A} \mathbf{w}$, where $\mathbf{A} \succ 0$ is a symmetric and positive definite matrix. The optimal solution occurs for $\mathbf{w}^* = 0$. When running gradient descent with constant step size $\eta > 0$, the obtained iterates would be $\mathbf{w}_k = (\mathbf{I} - \eta \mathbf{A}) \mathbf{w}_{k-1}$. This yields a convergence rate of $\frac{\|\mathbf{w}_k - \mathbf{w}^*\|}{\|\mathbf{w}_{k-1} - \mathbf{w}^*\|} \leq \max\left(|1 - \eta \lambda_{\max}(\mathbf{A})|, |1 - \eta \lambda_{\min}(\mathbf{A})|\right)$. The best convergence is obtained for $\eta = 2\left(\lambda_{\max}(\mathbf{A}) + \lambda_{\min}(\mathbf{A})\right)^{-1}$, resulting in $\|\mathbf{w}_k\| \leq \frac{\kappa - 1}{\kappa + 1} \|\mathbf{w}_{k-1}\|$, where $\kappa(\mathbf{A}) = \frac{\lambda_{\max}(\mathbf{A})}{\lambda_{\min}(\mathbf{A})}$ is called the condition number of the matrix. This ratio which, intuitively, measures how disparate are the largest and smallest curvature directions, is an indicator of the speed with which gradient descent would converge to the solution. When $\kappa \to \infty$, the progress can be painfully slow, and $\kappa \to 1$ indicates all the curvature directions are balanced, and thus the error along some direction does not trail behind the others, hence ensuring fast progress.

**Effect of condition number at initialization on the convergence rate**  As the condition number is a very local property, it is in general hard to connect the conditioning at network initialization to a global convergence rate. However, we would like to argue below that an ill-conditioned network initialization will still affect the rate of convergence for gradient descent (GD) in the initial phase of training. Let us denote the Lipschitz constant by $L$ and the smoothness constant by $\mu$. Furthermore, let the step size be such that $\eta_k \leq \frac{1}{L}$. We present a modified analysis of GD for strongly convex functions, where we use local constants $\mu(k)$ and $L(k)$ instead of the global smoothness and Lipschitz constant, respectively. Then by the definition of a single step of gradient descent and using the strong convexity and smoothness assumption[2] we have:

$$||\boldsymbol{\theta}_{k+1} - \boldsymbol{\theta}^*||^2 \leq (1 - \eta_k\mu)||\boldsymbol{\theta}_k - \boldsymbol{\theta}^*||^2 \tag{2}$$

So by recursively applying (2) and replacing $\mu$ by the local smoothness constants $\mu(k)$:

$$||\boldsymbol{\theta}_k - \boldsymbol{\theta}^*||^2 \leq \prod_{i=0}^{k-1}(1 - \eta_i\mu(i))||\boldsymbol{\theta}_0 - \boldsymbol{\theta}^*||^2 \tag{3}$$

One can clearly see the effect of $\mu(0)$ in the bound, which is even more dominant when $\mu(k)$ changes slowly. Of course, the effect of $\mu(0)$ attenuates over time, and that's why we are talking about a local effect. However, one should keep in mind that overparametrization leads the parameter to stay closer to initialization (at least in the NTK regime [Lee et al., 2019]).

## 3 Related work

Since the Gauss-Newton matrix is intimately related to the Hessian matrix, and the fact that towards the end of training, the Hessian approaches the Gauss-Newton matrix [Singh et al., 2021], we carry out a broader discussion of the related work, by including the significance of the Hessian at large.

**The relevance of the Hessian matrix for neural networks.**  (i) *Generalization-focused work:* There is a rich and growing body of work that points towards the significance of various Hessian-based measures in governing different aspects of optimization and generalization. One popular hypothesis [Hochreiter and Schmidhuber, 1997, Keskar et al., 2016, Dziugaite and Roy, 2017, Chaudhari et al., 2019] is that flatter minima generalize better, where the Hessian trace or the spectral norm is used to measure flatness. This hypothesis is not undisputed [Dinh et al., 2017], and the extent to which this is explanatory of generalization has been put to question recently [Granziol, 2021, Andriushchenko et al., 2023]. Nevertheless, yet another line of work has tried to develop regularization techniques that further encourage reaching a flatter minimum, as shown most prominently in proposing sharpness-aware minimization [Foret et al., 2021].

*(ii) Understanding Architectural and Training aspects of Neural Networks:* Some other work has studied the challenges in large-batch training via the Hessian spectrum in Yao et al. [2018]. Also, the use of large learning rates has been suggested to result in flatter minima via the initial catapult phase [Lewkowycz et al., 2020]. The effect of residual connections and Batch normalization on Hessian eigenspectrum were empirically studied in Ghorbani et al. [2019], Yao et al. [2020], which introduced PyHessian, a framework to compute Hessian information in a scalable manner. More recently, the so-called edge of stability [Cohen et al., 2021] phenomenon connects the optimization dynamics of gradient descent with the maximum eigenvalue of the loss Hessian. Very recently, the phenomenon of Deep neural collapse was studied in  Beaglehole et al. [2024] via the average gradient outer-product.

*(iii) Applications:* There has also been a dominant line of work utilizing the Hessian for second-order optimization, albeit via varying efficient approximations, most notably via K-FAC [Martens and Grosse, 2020] but also others such as Yao et al. [2021], Liu et al. [2023], Lin et al. [2023]. Given its versatile nature, the Hessian has also been used for model compression through pruning [LeCun et al., 1989, Hassibi et al., 1993, Singh and Alistarh, 2020] as well as quantization [Dong et al., 2019, Frantar et al., 2023], but also in understanding the sensitivity of predictions and the function learned by neural networks via influence functions [Koh and Liang, 2020, Grosse et al., 2023], and countless more.

**Theoretical and empirical studies of the Hessian.** There have been prior theoretical studies that aim to deliver an understanding of the Hessian spectrum in the asymptotic setting [Pennington and Bahri, 2017, Jacot et al., 2019], but it remains unclear how to extract results for finite-width

---

[2]For details of the derivation, we refer the reader to Appendix H.1

networks as used in practice. Besides, past work has analyzed the rank, empirically [Sagun et al., 2016, 2017] as well as theoretically [Singh et al., 2021, 2023]. Further, the layer-wise Hessian of a network can be roughly approximated by the Kronecker product of two smaller matrices, whose top eigenspace has shown to contain certain similarities with that of the Hessian [Wu et al., 2020]. In a different line of work by Liao and Mahoney [2021], the limiting eigenvalue distribution of the Hessian of generalized generalized linear models (G-GLMs) and the behaviour of potentially isolated eigenvalue-eigenvector pairs is analyzed.

**Hessian and landscape conditioning.** The stock of empirical repertoire in deep learning has been enriched by successful adaptive optimization methods plus their variants [Kingma and Ba, 2014] as well as various tricks of the trade, such as Batch Normalization [Ioffe and Szegedy, 2015], Layer Normalization [Ba et al., 2016], orthogonal initialization [Saxe et al., 2013, Hu et al., 2020], and the kind, all of which can arguably be said to aid the otherwise ill-conditioning of the landscape. There have also been theoretical works establishing a link between the conditioning of the Hessian, at the optimum and the double-descent like generalization behavior of deep networks [Belkin et al., 2019, Singh et al., 2022].

**Gauss-Newton matrix and NTK.** In the context of over-parametrized networks, $\mathbf{G}_O$ is for instance connected to the (empirical) Neural Tangent Kernel, which has been the focus of a major line of research in the past few years [Jacot et al., 2018, Wang et al., 2022, Yang, 2020] as the NTK presents an interesting limit of infinite-width networks. As a result the asymptotic spectrum and the minimum eigenvalue of the NTK has been studied in [Nguyen et al., 2022, Liu and Hui, 2023], but the implications for finite-width networks remain unclear.

Despite this and the other extensive work discussed above, a detailed theoretical study on the Gauss Newton conditioning of neural networks has been absent. In particular, there has been little work trying to understand the precise sources of ill-conditioning present within deep networks. Therefore, here we try to dissect the nature of conditioning itself, via a first principle approach. We hope this will spark further work that aims to precisely get to the source of ill-conditioning in neural networks and, in a longer horizon, helps towards designing theoretically-guided initialization strategies or normalization techniques that seek to also ensure better conditioning of the GN matrix.

# 4 Theoretical characterisation

The main part of this paper will focus on analyzing the conditioning of $\mathbf{G}_O$ in Eq. (1) as prior work [Ren and Goldfarb, 2019, Schraudolph, 2002] has demonstrated its heightened significance in influencing the optimization process. We will further discuss an extension to $\mathbf{H}_F$ in Appendix F. Tying both bounds together yields a bound on the condition number of the overall loss Hessian in some simple setups.

**Pseudo-condition number.** Since the GN matrix of deep networks is not necessarily full rank [Sagun et al., 2017, Singh et al., 2021], we will analyze the pseudo-condition number defined as the ratio of the maximum eigenvalue over the minimum *non-zero* eigenvalue. This choice is rationalized by the fact that gradient-based methods will effectively not steer into the GN null space, and we are interested in the conditioning of the space in which optimization actually proceeds. For brevity, we will skip making this distinction between condition number and pseudo-condition number hereafter.

## 4.1 Spectrum of Gauss-Newton matrix

We will start with the case of linear activations. In this case a network with $L$ layers can be expressed by $\mathbf{F}_{\boldsymbol{\theta}}(\mathbf{x}) = \mathbf{W}^L \mathbf{W}^{L-1} \cdots \mathbf{W}^1 \mathbf{x}$, with $\mathbf{W}^\ell \in \mathbb{R}^{a_\ell \times a_{\ell-1}}$ for $\ell = 1, \ldots, L$ and $a_L = k, a_0 = d$. To facilitate the presentation of the empirical work, we will assume that the widths of all hidden layers are the same, i.e. $\alpha_\ell = m$ for all $\ell = 1, \ldots, L-1$. Also, let us denote by $\boldsymbol{\Sigma} = \frac{1}{n} \sum_{i=1}^n \mathbf{x}_i \mathbf{x}_i^T \in \mathbb{R}^{d \times d}$ the empirical input covariance matrix. Furthermore, we introduce the shorthand-notation $\mathbf{W}^{k:\ell} = \mathbf{W}^k \cdots \mathbf{W}^\ell$ for $k > \ell$ and $k < \ell$, $\mathbf{W}^{k:\ell} = \mathbf{W}^{k^\top} \cdots \mathbf{W}^{\ell^\top}$. Then, Singh et al. [2021] show that the GN matrix can be decomposed as $\mathbf{G}_O = \mathbf{U}(\mathbf{I}_k \otimes \boldsymbol{\Sigma})\mathbf{U}^\top$, where $\mathbf{I}_k$ is the identity matrix of dimension $k$ and $\mathbf{U} \in \mathbb{R}^{p \times kd}$ is given by:

$$\mathbf{U} = \left( \mathbf{W}^{2:L} \otimes \mathbf{I}_d \ \ldots \ \mathbf{W}^{\ell+1:L} \otimes \mathbf{W}^{\ell-1:1} \ \ldots \ \mathbf{I}_k \otimes \mathbf{W}^{L-1:1} \right)^\top .$$

By rewriting $\mathbf{U}(\mathbf{I}_K \otimes \boldsymbol{\Sigma})\mathbf{U}^\top = \mathbf{U}(\mathbf{I}_K \otimes \boldsymbol{\Sigma}^{1/2})(\mathbf{I}_K \otimes \boldsymbol{\Sigma}^{1/2})\mathbf{U}^\top$, where $\boldsymbol{\Sigma}^{1/2}$ is the unique positive semi-definite square root of $\boldsymbol{\Sigma}$ and noting that $\mathbf{AB}$ and $\mathbf{BA}$ have the same non-zero eigenvalues, we

have that the non-zero eigenvalues of $\mathbf{G}_O$ are the same as those of

$$\widehat{\mathbf{G}}_O = (\mathbf{I}_K \otimes \boldsymbol{\Sigma}^{1/2})\mathbf{U}^\top \mathbf{U}(\mathbf{I}_K \otimes \boldsymbol{\Sigma}^{1/2}), \tag{4}$$

where $\mathbf{U}^\top \mathbf{U} \in \mathbb{R}^{Kd \times Kd}$ is equal to

$$\mathbf{U}^\top \mathbf{U} = \sum_{\ell=1}^{L} \mathbf{W}^{L:\ell+1}\mathbf{W}^{\ell+1:L} \otimes \mathbf{W}^{1:\ell-1}\mathbf{W}^{\ell-1:1}. \tag{5}$$

**Warm-up: the one-hidden layer case.** In the case of one-hidden layer network, $\mathrm{F}_{\boldsymbol{\theta}}(\mathbf{x}) = \mathbf{W}\mathbf{V}\mathbf{x}$, we have that $\widehat{\mathbf{G}}_O = \mathbf{W}\mathbf{W}^\top \otimes \boldsymbol{\Sigma} + \mathbf{I}_k \otimes \boldsymbol{\Sigma}^{1/2}\mathbf{V}^\top \mathbf{V}\boldsymbol{\Sigma}^{1/2}$. To derive an upper bound of the condition number, we will lower bound the smallest eigenvalue $\lambda_{\min}(\widehat{\mathbf{G}}_O)$ and upper bound the largest eigenvalue $\lambda_{\max}(\widehat{\mathbf{G}}_O)$ separately. Using standard perturbation bounds for matrix eigenvalues discussed in Appendix A, we obtain the following upper bound on the condition number:

**Lemma 1.** *Let $\beta_w = \sigma_{\min}^2(\mathbf{W})/(\sigma_{\min}^2(\mathbf{W}) + \sigma_{\min}^2(\mathbf{V}))$. Then the condition number of GN for the one-hidden layer network with linear activations with $m > \max\{d, k\}$ is upper bounded by*

$$\kappa(\widehat{\mathbf{G}}_O) \leq \kappa(\boldsymbol{\Sigma}) \cdot \frac{\sigma_{\max}^2(\mathbf{W}) + \sigma_{\max}^2(\mathbf{V})}{\sigma_{\min}^2(\mathbf{W}) + \sigma_{\min}^2(\mathbf{V})} = \kappa(\boldsymbol{\Sigma}) \cdot (\beta_w \, \kappa(\mathbf{W})^2 + (1 - \beta_w) \, \kappa(\mathbf{V})^2). \tag{6}$$

*Proof sketch.* Choosing $\mathbf{A} = \mathbf{W}\mathbf{W}^\top \otimes \boldsymbol{\Sigma}$ and $\mathbf{B} = \mathbf{I}_k \otimes \boldsymbol{\Sigma}^{1/2}\mathbf{V}^\top \mathbf{V}\boldsymbol{\Sigma}^{1/2}$ and $i = j = 1$ for the Weyl's inequality, $i = j = n$ for the dual Weyl's inequality and using the fact that $\mathbf{W}\mathbf{W}^\top \otimes \boldsymbol{\Sigma}$ and $\mathbf{I}_k \otimes \boldsymbol{\Sigma}^{1/2}\mathbf{V}^\top \mathbf{V}\boldsymbol{\Sigma}^{1/2}$ are positive semidefinite yield the result. $\square$

It is important to point out that the convex combination in Eq. (6) is crucial and a more naive bound where we take the maximum of both terms is instead too loose, see details in Appendix E. Later on, we will observe that the general case with $L$ layers also exhibits a comparable structure, wherein the significance of the convex combination becomes even more pronounced in deriving practical bounds.

**Remark R2** (Role of the data covariance). *Besides the above dependence in terms of the convex combination of the bounds, we also see how the conditioning of the input data affects the conditioning of the GN spectra. This is observed in Figure 19, where the condition number of the GN matrix is calculated on whitened and not whitened data. This observation might also shed light on why data normalization remains a standard choice in deep learning, often complemented by the normalization of intermediate layer activations.*

## 4.2 The general $L$-layer case

Following our examination of the single-hidden layer case, we now broaden our analysis to include $L$-layer linear networks. As before, we will first derive an expression of the GN matrix and subsequently bound the largest and smallest eigenvalue separately to derive a bound on the condition number. Obtaining the GN matrix involves combining (4) and (5), which yields

$$\widehat{\mathbf{G}}_O = (\mathbf{I}_K \otimes \boldsymbol{\Sigma}^{1/2})\mathbf{U}^\top \mathbf{U}(\mathbf{I}_K \otimes \boldsymbol{\Sigma}^{1/2}) = \sum_{l=1}^{L} (\mathbf{W}^{L:\ell+1}\mathbf{W}^{\ell+1:L}) \otimes (\boldsymbol{\Sigma}^{1/2}\mathbf{W}^{1:\ell-1}\mathbf{W}^{\ell-1:1}\boldsymbol{\Sigma}^{1/2}).$$

By repeatedly applying Weyl's inequalities, we obtain an upper bound on the condition number.

**Lemma 2.** *Assume that $m > \max\{d, k\}$ and that $\alpha_\ell := \sigma_{\min}^2(\mathbf{W}^{L:\ell+1}) \cdot \sigma_{\min}^2(\mathbf{W}^{1:\ell-1}) > 0 \ \forall \ell = 1, \ldots, L$. Let $\gamma_\ell := \frac{\alpha_\ell}{\sum_{i=1}^{L} \alpha_i}$. Then the condition number of the GN matrix of a $L$-layer linear network can be upper-bounded in the following way:*

$$\kappa(\widehat{\mathbf{G}}_O) \leq \kappa(\boldsymbol{\Sigma}) \sum_{\ell=1}^{L} \gamma_\ell \kappa(\mathbf{W}^{L:\ell+1})^2 \kappa(\mathbf{W}^{1:\ell-1})^2 \leq \kappa(\boldsymbol{\Sigma}) \max_{1 \leq \ell \leq L} \left\{ \kappa(\mathbf{W}^{L:\ell+1})^2 \kappa(\mathbf{W}^{1:\ell-1})^2 \right\}.$$

As mentioned earlier, we observe the same convex combination structure which, as we will soon see experimentally, is crucial to obtain a bound that works in practice.

**Empirical validation.** The empirical results in Figure 2a show that the derived bound seems to be tight and predictive of the trend of the condition number of GN at initialization. If the width of the hidden layer is held constant, the condition number grows with a quadratic trend. However, the condition number can be controlled if the width is scaled proportionally with the depth. This gives another explanation of why in practice the width of the network layers is scaled proportionally with the depth to enable faster network training.

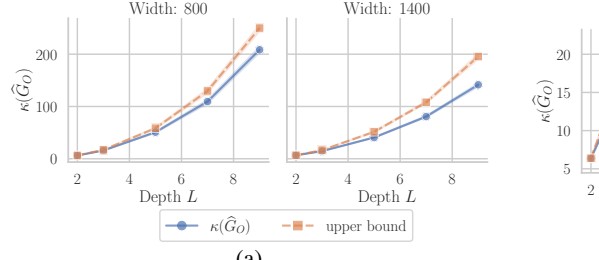 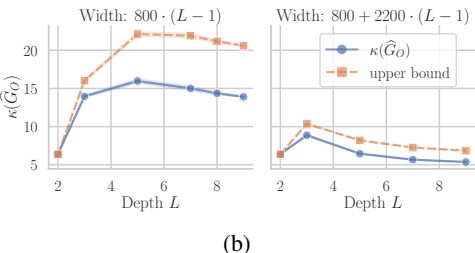

(a)   (b)

Figure 2: **a)** Condition number at initialization under Kaiming normal initialization of GN $\kappa(\hat{\mathbf{G}}_O)$ and first upper bound derived in Lemma 2 and Eq.(7) for whitened MNIST as a function of depth $L$ for different hidden layer widths $m$ for a Linear Network over 3 initializations. **b)** Scaling the width of the hidden layer proportionally to the depth leads to slower growth of the condition number (left) or improves the condition number with depth if the scaling factor is chosen sufficiently large (right).

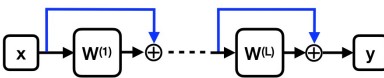 Figure 4: Adding skip connections between each layer for a general L-layer linear Neural Network.

Furthermore, Figure 3 empirically demonstrates the importance of bounding the condition number as a convex combination of the condition number of the weight matrices, as simply taking the maximum makes the bound vacuous. This is due to a 'self-balancing' behavior of the individual terms in Lemma 2, which is further elaborated in Appendix E. We show that the difference in magnitude between each condition number is largely driven by their smallest eigenvalue. Thus at the same time, they also have a smaller weight in the convex combination, making the overall condition number grow slower than the maximum bound would predict.

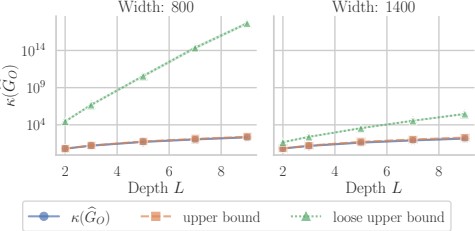

Figure 3: Comparison of derived upper bounds in Lemma 2 for the condition number at initialization for whitened MNIST over 20 runs. Note the logarithmic scaling of the $y$-axis.

### 4.3   $L$-**layer linear residual networks**

Following the examination of linear networks, our focus will now shift to the analysis of linear networks where we introduce a skip connection with weight $\beta$ between each layer, as illustrated in Figure 4. This results in a residual network of the form $F_\theta(\mathbf{x}) = (\mathbf{W}^{(L)} + \beta\mathbf{I})\cdots(\mathbf{W}^{(1)} + \beta\mathbf{I})\mathbf{x}$. Given the architecture of the residual network, a substantial portion of the analysis conducted in Section 4.2 for the $L$-layer linear neural network can be repurposed.

The key insight lies in the realization that when the skip connection precisely bypasses one layer, it implies a modification of Eq. (5) to $\mathbf{U}^\top\mathbf{U} = \sum_{\ell=1}^{L} \mathbf{W}_\beta^{L:\ell+1}\mathbf{W}_\beta^{\ell+1:L} \otimes \mathbf{W}_\beta^{1:\ell-1}\mathbf{W}_\beta^{\ell-1:1}$, where we define $\mathbf{W}_\beta^{k:l} := (\mathbf{W}^k + \beta\mathbf{I})\cdots(\mathbf{W}^l + \beta\mathbf{I})$ if $k > l$ and $\mathbf{W}_\beta^{k:l} := (\mathbf{W}^k + \beta\mathbf{I})^\top \cdots (\mathbf{W}^l + \beta\mathbf{I})^\top$ if $k < l$. $\mathbf{I}$ denotes the rectangular identity matrix of appropriate dimensions. Note, that we can apply the upper bounds derived in Lemma 2 analogously to arrive at

$$\kappa(\hat{\mathbf{G}}_O) \le \kappa(\mathbf{\Sigma}) \sum_{\ell=1}^{L} \gamma_\ell^\beta \, \kappa^2(\mathbf{W}_\beta^{L:\ell+1}) \, \kappa^2(\mathbf{W}_\beta^{1:\ell-1}) \le \kappa(\mathbf{\Sigma}) \max_{1 \le \ell \le L} \kappa^2(\mathbf{W}_\beta^{L:\ell+1}) \, \kappa^2(\mathbf{W}_\beta^{1:\ell-1}) \quad (7)$$

where $\gamma_\ell^\beta := \frac{\alpha_\ell^\beta}{\sum_{i=1}^{L} \alpha_i^\beta}$ and analogously to the $L$-layer case $\alpha_\ell^\beta := \sigma_{\min}^2(\mathbf{W}_\beta^{L:\ell+1}) \cdot \sigma_{\min}^2(\mathbf{W}_\beta^{1:\ell-1})$ for $\ell = 1, \dots, L$.

Let us now analyze how skip connections affect the condition number. Denoting the SVD of $\mathbf{W}^\ell = \mathbf{U}^\ell \mathbf{S}^\ell \mathbf{V}^{\ell\top}$ note that $\mathbf{W}^\ell + \beta\mathbf{I} = \mathbf{U}^\ell(\mathbf{S}^\ell + \beta\mathbf{I})\mathbf{V}^{\ell T}$, and therefore $\sigma_i(\mathbf{W}^\ell + \beta\mathbf{I}) = \sigma_i(\mathbf{W}^\ell) + \beta$ for all singular values of $\mathbf{W}^\ell$. Incorporating skip connections results in a spectral shift of each weight matrix to the positive direction by a magnitude of $\beta$. Furthermore, we will assume that the left and right

singular vectors of the layers in-between coincide, that is $\mathbf{V}^\ell = \mathbf{U}^{\ell-1}$. This assumption is fulfilled if the initialization values are sufficiently close to zero [Saxe et al., 2013]. Then for each $\ell$ we have

$$\kappa^2(\mathbf{W}_\beta^{L:\ell+1})\kappa^2(\mathbf{W}_\beta^{1:\ell-1}) = \frac{\prod_{i=\ell+1}^L(\sigma_{\max}(\mathbf{W}^i)+\beta)^2 \prod_{i=1}^{\ell-1}(\sigma_{\max}(\mathbf{W}^i)+\beta)^2}{\prod_{i=\ell+1}^L(\sigma_{\min}(\mathbf{W}^i)+\beta)^2 \prod_{i=1}^{\ell-1}(\sigma_{\min}(\mathbf{W}^i)+\beta)^2}$$

$$= \prod_{i\neq\ell}\left(\frac{\sigma_{\max}(\mathbf{W}^i)+\beta}{\sigma_{\min}(\mathbf{W}^i)+\beta}\right)^2. \tag{8}$$

Since $\frac{\sigma_{\max}(\mathbf{W}^i)+\beta}{\sigma_{\min}(\mathbf{W}^i)+\beta} < \frac{\sigma_{\max}(\mathbf{W}^i)}{\sigma_{\min}(\mathbf{W}^i)}$ for all $\beta > 0$, we can conclude that the second upper bound in Eq. (7) for residual networks is smaller than the second upper bound for linear networks in Lemma 2. We verify that this observation also holds for the actual condition number and the tight upper bound in Eq. (7) as can be seen in Figure 5. The latter shows that adding skip connections improves the conditioning of the network. Furthermore, Figure 20 in the appendix shows that the condition number is improved more for larger $\beta$ because this pushes the condition number towards one as can be seen from Eq. (8). To conclude, the analysis of the condition number provides a rigorous explanation for the popularity of residual networks compared to fully connected networks. This is in line with existing empirical results in the literature [He et al., 2016, Li et al., 2018, Liu et al., 2019].

**Remark: extension to convolutional layers.** Note that the analysis on fully connected layers can also be extended to convolutional layers in a straightforward way by making use of the fact that the convolution operation can be reformulated in the form of a matrix-vector product using Toeplitz matrices, as discussed for instance in Singh et al. [2023]. In this case, we can apply the same analysis as for fully connected networks. See Appendix B for more details.

## 5 Extension to non-linear activations

After the analysis of networks with a linear activation in the previous section, we will extend our results to non-linear activation functions $\sigma$ that satisfy $\sigma(z) = \sigma'(z)z$ for one-hidden layer networks $F_\theta(\mathbf{x}) = \mathbf{W}\sigma(\mathbf{V}\mathbf{x})$. Specifically, this extension includes all piece-wise linear activation functions such as ReLU or Leaky ReLU. For this purpose, we first rewrite the network output as a super-position of unit-networks $F_{\theta_i}(\mathbf{x})$ over the number of hidden neurons $m$ as

$$F_\theta(\mathbf{x}) = \sum_{i=1}^m F_{\theta_i}(\mathbf{x}) = \sum_{i=1}^m \mathbf{W}_{\bullet,i}\sigma(\mathbf{V}_{i,\bullet}\mathbf{x}), \tag{9}$$

where $\mathbf{W}_{\bullet,i}$ denotes the $i$-th column of $\mathbf{W}$ and $\mathbf{V}_{i,\bullet}$ denotes the $i$-th row of $\mathbf{V}$. Using the following lemma, we can then derive an expression for the Gauss-Newton matrix.

**Lemma 3.** *(Lemma 25 in Singh et al. [2021]) Let $F_{\theta_i}(\mathbf{x}) = \mathbf{W}_{\bullet,i}\sigma(\mathbf{V}_{i,\bullet}\mathbf{x})$ be a unit-network corresponding to $i$-th neuron, with the non-linearity $\sigma$ such that $\sigma(z) = \sigma'(z)z$. Let $\mathbf{X} \in \mathbb{R}^{d\times n}$ denote the data matrix. Further, let $\mathbf{\Lambda}^i \in \mathbb{R}^{n\times n}$ be defined as $(\mathbf{\Lambda}^i)_{jj} = \sigma'(\mathbf{V}_{i,\bullet}\mathbf{x})_j$, for $j = 1,\ldots,n$ and zero elsewhere. Then the Jacobian matrix $\nabla_\theta F_{\theta_i}(\mathbf{X})$ is given (in transposed form) by:*

$$\nabla_\theta F_{\theta_i}(\mathbf{X}) = \begin{pmatrix} \mathbf{0} & \mathbf{X}\mathbf{\Lambda}^i \otimes \mathbf{W}_{\bullet,i}^\top & \mathbf{V}_{i,\bullet}\mathbf{X}\mathbf{\Lambda}^i \otimes \mathbf{I}_k & \mathbf{0} \end{pmatrix}^\top.$$

The Jacobian matrix of the full network is simply the sum over all units $\nabla_\theta F_\theta(\mathbf{X}) = \sum_{i=1}^m \nabla_\theta F_{\theta_i}(\mathbf{X})$. Now that we have an expression for the Jacobian matrix, we can compute the GN matrix as $\mathbf{G}_O = \nabla F_\theta(\mathbf{X})\nabla F_\theta(\mathbf{X})^\top$. Note that the non-zero eigenvalue of $\mathbf{G}_O$ and $\widehat{\mathbf{G}}_O := \nabla F_\theta(\mathbf{X})^\top \nabla F_\theta(\mathbf{X}) \in \mathbb{R}^{kn\times kn}$ are the same, for which we have

$$\widehat{\mathbf{G}}_O = \sum_{i=1}^m \nabla F_{\theta_i}(\mathbf{X})^\top \nabla F_{\theta_i}(\mathbf{X}) = \sum_{i=1}^m \widehat{\mathbf{G}}_O^i$$

with $\widehat{\mathbf{G}}_O^i := \mathbf{\Lambda}^i\mathbf{X}^\top\mathbf{X}\mathbf{\Lambda}^i \otimes \mathbf{W}_{\bullet i}\mathbf{W}_{\bullet i}^\top + \mathbf{\Lambda}^i\mathbf{X}^\top\mathbf{V}_{i\bullet}^\top\mathbf{V}_{i\bullet}\mathbf{X}\mathbf{\Lambda}^i \otimes \mathbf{I}_k \in \mathbb{R}^{kn\times kn}$, since the mixed terms $\nabla F_{\theta_i}(\mathbf{X})^\top \nabla F_{\theta_j}(\mathbf{X})$ with $i \neq j$ are zero due to the block structure of the Jacobian of the unit-networks. Using the mixed product property of the Kronecker product we obtain

$$\widehat{\mathbf{G}}_O = \sum_{i=1}^m \mathbf{\Lambda}^i\mathbf{X}^\top\mathbf{X}\mathbf{\Lambda}^i \otimes \mathbf{W}_{\bullet i}\mathbf{W}_{\bullet i}^\top + \sum_{i=1}^m \mathbf{\Lambda}^i\mathbf{X}^\top\mathbf{V}_{i\bullet}^\top\mathbf{V}_{i\bullet}\mathbf{X}\mathbf{\Lambda}^i \otimes \mathbf{I}_k. \tag{10}$$

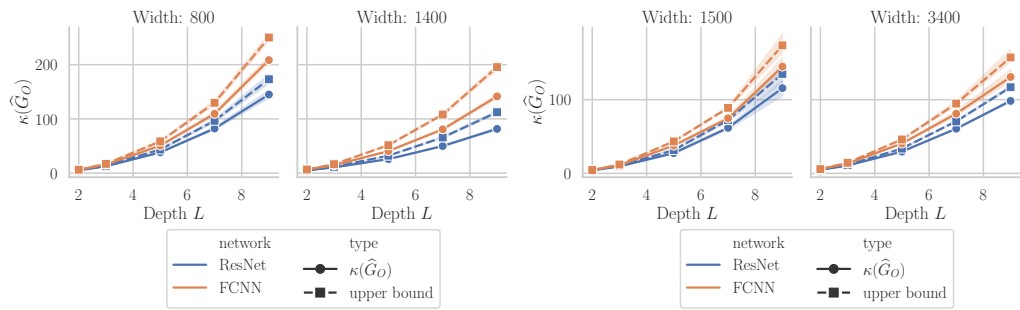

Figure 5: Comparison of condition number of GN $\kappa(\widehat{\mathbf{G}}_O)$ between Linear Network and Residual Network with $\beta = 1$ for whitened MNIST (left) and whitened Cifar-10 (right) at initialization using Kaiming normal initialization over three seeds. The upper bounds refer to the first upper bound in Lemma 2 and Eq. (7), respectively.

To improve readability, we will write $\boldsymbol{\Gamma} := \sum_{i=1}^{m} \boldsymbol{\Lambda}^i \mathbf{X}^\top \mathbf{V}_{i\bullet}^\top \mathbf{V}_{i\bullet} \mathbf{X} \boldsymbol{\Lambda}^i$ hereafter.

**Leaky ReLU activation.** Let's now consider the case where the non-linear activation $\sigma$ is LeakyReLU$(x) = \max\{\alpha x, x\}$ for some constant $\alpha \in [0, 1)$. Typically $\alpha$ is chosen to be close to zero, e.g. $\alpha = 0.01$. Again, the condition number $\kappa(\widehat{\mathbf{G}}_O)$ can be upper bounded by bounding the extreme eigenvalues independently. By observing that all terms in Eq. (10) are symmetric, we can again apply the Weyl's and dual Weyl's inequality. However to get a lower bound larger than zero for $\lambda_{\min}(\widehat{\mathbf{G}}_O)$ a different approach is needed as $\mathbf{W}_{\bullet,i} \mathbf{W}_{\bullet,i}^\top$ is rank-deficient and the same steps to bound the smallest eigenvalue would lead to a vacuous value of zero. Instead, we will use the following observation that we can bound the extreme eigenvalues of the sum of a Kronecker product of PSD matrices $\mathbf{A}_i, \mathbf{B}_i$ for $i = 1, \ldots, m$ by $\lambda_{\min}\left(\sum_{i=1}^{m}(\mathbf{A}_i \otimes \mathbf{B}_i)\right) \geq \min_{1 \leq j \leq m} \lambda_{\min}(\mathbf{A}_j) \cdot \lambda_{\min}\left(\sum_{i=1}^{m} \mathbf{B}_i\right)$.

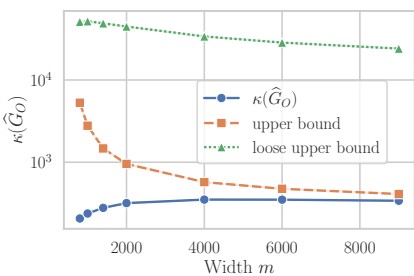

Figure 6: Condition number of a one-hidden layer Leaky ReLU network with $\alpha = 0.01$ and upper bounds for whitened MNIST over $n = 500$ data points. The upper bound refers to Eq. (11). Note that the y-axis is log scaled.

By first using Weyl's inequalities and subsequently applying the above bound with $\mathbf{A}_i = \boldsymbol{\Lambda}^i \mathbf{X}^\top \mathbf{X} \boldsymbol{\Lambda}^i$ and $\mathbf{B}_i = \mathbf{W}_{\bullet,i} \mathbf{W}_{\bullet,i}^\top$ we achieve the following bound for the condition number for $\kappa(\widehat{\mathbf{G}}_O)$.

**Lemma 4.** *Consider an one-hidden layer network with a Leaky-ReLU activation with negative slope $\alpha \in [0, 1]$. Then the condition number of the GN matrix defined in Equation* (10) *can be bounded as:*

$$\kappa(\widehat{\mathbf{G}}_O) \leq \frac{\sigma_{\max}^2(\mathbf{X})\,\sigma_{\max}^2(\mathbf{W}) + \lambda_{\max}(\boldsymbol{\Gamma})}{\alpha^2\,\sigma_{\min}^2(\mathbf{X})\,\sigma_{\min}^2(\mathbf{W}) + \lambda_{\min}(\boldsymbol{\Gamma})} \quad (11)$$

The proof can be found in Appendix H. We further empirically validate the tightness of the upper bounds on a subset of the MNIST dataset ($n = 500$), where we chose $\alpha = 0.01$, which is the default value in Pytorch [Paszke et al., 2019]. As can be seen in Figure 6, contrary to the linear setting the condition number increases with width. The upper bound also becomes tighter and more predictive with increasing width.

**Comparison to linear network.** By running the same setup for varying values of $\alpha \in [0, 1]$ we can interpolate between ReLU ($\alpha = 0$) and linear activation ($\alpha = 1$) to see how the introduction of the non-linearity affects the conditioning. In Figure 23, which had to be deferred to the appendix due to space constraints, we can see how reducing the value of $\alpha$ seems to consistently improve the conditioning of the GN matrix for the one-hidden layer case.

## 6  Conditioning under batch normalization

Based on the observation that conditioning of the input data affects the conditioning of the GN spectrum, we also tested whether Batch normalization (BN) [Ioffe and Szegedy, 2015], which is a very commonly used normalization scheme, has a similar effect on the condition number. For this, the condition number of the GN matrix was calculated for a one-hidden layer linear network with and without a Batch normalization layer. The experiment was run on a downsampled and subsampled version of Cifar-10, which was converted to grayscale with $d = 64$ and $n = 1000$. The data was not whitened to see the effect of BN more strongly. As can be seen in Figure 7, we indeed observe a clear improvement of the condition number when Batch normalization layers are added. Also, the trend of improved conditioning with increasing width remains after adding BN.

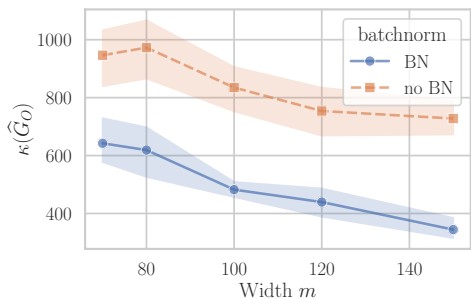

Figure 7: Comparison of conditioning of the GN matrix for a one-hidden layer linear network with and without Batch normalization on downsampled Cifar-10 data ($d = 64, n = 1000$) at initialization over 5 runs.

## 7  Discussion and conclusion

**Summary.** In this work, we derived new analytical bounds for the condition number of neural networks, showcasing the following key findings: **a)** The conditioning of the input data has a nearly proportional impact on the conditioning of the GN spectra, underscoring the significance of data normalization. Empirical evidence further demonstrates that Batch Normalization similarly enhances the condition number. **b)** For linear networks, we showed that the condition number grows quadratically with depth for fixed hidden width. Also, widening hidden layers improves conditioning, and scaling the hidden width proportionally with depth can compensate for the growth. **c)** We showed how adding residual connections improves the condition number, which also explains how they enable the training of very deep networks. **d)** Preliminary experiments suggest that the ReLU activation seems to improve the conditioning compared to linear networks in the one-hidden layer case.

**Interesting use cases of our results.** Through our analysis, we highlighted that the condition number as a tool from classical optimization is also an attractive option to better understand challenges in neural network training with gradient-based methods. Especially, knowing how different architectural choices will affect the optimization landscape provides a more principled way to design the network architecture for practitioners, for instance how to scale the width in relation to the depth of the network. The paper also gives a justification of why pruned networks are more difficult to train as they have worse conditioning. Although this is not our focus, it is possible that our analysis could inspire better techniques for pruning neural networks.

**Limitations and future work.** We made a first step toward understanding the impact of different architectural choices on the conditioning of the optimization landscape. However, there are still many design choices, that have not been covered yet, such as an extension to non-linear networks for arbitrary layers, other architectures, such as transformers, and analytic bounds for normalization schemes, such as batch or layer normalization, which we will leave to future work. Another limitation of our current work is that the derived upper bounds are agnostic to the training dynamics. Therefore, they cannot distinguish the difference between random initializations and solutions of deep learning models after convergence. Incorporating training dynamics into the upper bounds would allow us to characterize solutions found by different architectures, which is left for future work. Another future direction is to extend the analysis to the Generalized Gauss-Newton matrix, which is particularly relevant for training with cross-entropy loss.

## Acknowledgements

Aurelien Lucchi acknowledges the financial support of the Swiss National Foundation, SNF grant No 223031. Sidak Pal Singh would like to acknowledge the financial support of Max Planck ETH Center for Learning Systems.

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

# A Standard perturbation bounds for matrix eigenvalues

## A.1 Weyl's inequality and dual Weyl's inequality

In this section, we first briefly review the Weyl's inequality and its dual form. We then demonstrate how these inequalities can be used to bound the extreme eigenvalues and thus the condition number of the sum of two Hermitian matrices $\mathbf{A}, \mathbf{B}$ by the extreme eigenvalues of $\mathbf{A}$ and $\mathbf{B}$.

**Lemma 5.** *Weyl [1912] Let $\mathbf{A}, \mathbf{B} \in \mathbb{R}^{n \times n}$ be two Hermitian matrices, and let $\lambda_1 \geq \ldots \geq \lambda_n$ denote the ordered eigenvalues, then the Weyl's inequality and dual Weyl's inequality state that*

$$\lambda_{i+j-1}(\mathbf{A} + \mathbf{B}) \leq \lambda_i(\mathbf{A}) + \lambda_j(\mathbf{B}), \ i, j \geq 1, i + j - 1 \leq n \qquad \text{(Weyl)}$$
$$\lambda_{i+j-n}(\mathbf{A} + \mathbf{B}) \geq \lambda_i(\mathbf{A}) + \lambda_j(\mathbf{B}), \ i, j \geq 1, i + j - n \leq n. \qquad \text{(Dual Weyl)}$$

In the following, we derive an upper and lower bound of the condition number of a symmetric matrix $\mathbf{A}$, for which we have

$$\kappa(\mathbf{A}) = \frac{|\lambda_{\max}(\mathbf{A})|}{|\lambda_{\min}(\mathbf{A})|}.$$

Recall that given a Hermitian $n \times n$ matrix $\mathbf{A}$, we can diagonalize it by the spectral theorem. We obtain the following sequence of real eigenvalues

$$\lambda_1(\mathbf{A}) \geq \lambda_2(\mathbf{A}) \geq \ldots \geq \lambda_n(\mathbf{A}).$$

We will make use of the Weyl's inequality and the dual Weyl's inequality to bound the eigenvalues of the sum of two Hermitian matrices $\mathbf{A} + \mathbf{B}$.

From Lemma 5, we obtain the following inequalities: From Weyl's inequality we have

$$\lambda_1(\mathbf{A} + \mathbf{B}) \leq \lambda_1(\mathbf{A}) + \lambda_1(\mathbf{B})$$
$$\lambda_n(\mathbf{A} + \mathbf{B}) \leq \lambda_{n-j}(\mathbf{A}) + \lambda_{1+j}(\mathbf{B}), \quad \text{for } 0 \leq j \leq n - 1.$$

From the dual Weyl's inequality we have

$$\lambda_1(\mathbf{A} + \mathbf{B}) \geq \lambda_{n-j}(\mathbf{A}) + \lambda_{1+j}(\mathbf{B}), \quad \text{for } 0 \leq j \leq n - 1$$
$$\lambda_n(\mathbf{A} + \mathbf{B}) \geq \lambda_n(\mathbf{A}) + \lambda_n(\mathbf{B}).$$

Hence, we can find a lower and upper bound for the condition number of $\mathbf{A} + \mathbf{B}$

$$\frac{|\max_{0 \leq j \leq n-1}(\lambda_{n-j}(\mathbf{A}) + \lambda_{1+j}(\mathbf{B}))|}{|\min_{0 \leq j \leq n-1}(\lambda_{n-j}(\mathbf{A}) + \lambda_{1+j}(\mathbf{B}))|} \leq \kappa(\mathbf{A} + \mathbf{B}) \leq \frac{|\lambda_{\max}(\mathbf{A}) + \lambda_{\max}(\mathbf{B})|}{|\lambda_{\min}(\mathbf{A}) + \lambda_{\min}(\mathbf{B})|}. \qquad (12)$$

## A.2 Further bounds for extreme eigenvalues for positive semidefinite matrices

For square matrices $\mathbf{A} \in \mathbb{R}^{p \times p}, \mathbf{B} \in \mathbb{R}^{q \times q}$, denote by $\lambda_1, \ldots, \lambda_p$ the eigenvalues of $\mathbf{A}$ and by $\mu_1, \ldots, \mu_q$ the eigenvalues of $\mathbf{B}$ listed by multiplicity. Then the eigenspectrum of $\mathbf{A} \otimes \mathbf{B} \in \mathbb{R}^{pq \times pq}$ consists of the eigenvalues

$$\lambda_i \mu_j \quad i = 1, \ldots, p, \ j = 1, \ldots, q.$$

Note that if $\mathbf{A}, \mathbf{B}$ are additionally positive semidefinite (PSD), we have the following equality for the largest and smallest eigenvalues of $\mathbf{A} \otimes \mathbf{B}$

$$\lambda_{\max}(\mathbf{A} \otimes \mathbf{B}) = \lambda_{\max}(\mathbf{A})\lambda_{\max}(\mathbf{B}) \qquad (13)$$
$$\lambda_{\min}(\mathbf{A} \otimes \mathbf{B}) = \lambda_{\min}(\mathbf{A})\lambda_{\min}(\mathbf{B}). \qquad (14)$$

Furthermore, it follows from the sub-multiplicativity of the matrix norm that for square, PSD matrices $\mathbf{A}, \mathbf{B}$ of same dimensions it holds that

$$\lambda_{\max}(\mathbf{AB}) \leq \lambda_{\max}(\mathbf{A}) \cdot \lambda_{\max}(\mathbf{B})$$
$$\lambda_{\min}(\mathbf{AB}) \geq \lambda_{\min}(\mathbf{A}) \cdot \lambda_{\min}(\mathbf{B}).$$

# B Extension of analysis to convolutional layers

The analysis on fully connected layers can be extended to convolutional layers in a straight forward way, by using the fact that the convolution operation can be reformulated in the form of a matrix-vector product using Toeplitz matrices, as discussed for instance in Singh et al. [2023]. This would mean that we can apply the same analysis as for fully connected networks. For completeness we will discuss how below how the convolution operation can be rewritten as a matrix-vector product. For further details, the readers is referred to Singh et al. [2023].

Consider a convolution operation $\mathbf{W} * \mathbf{x}$ of an input $\mathbf{x} \in \mathbb{R}^d$ with $m$ filters of size $k \leq d$, which are organized in the matrix $\mathbf{W} \in \mathbb{R}^{m \times k}$. For simplicity a stride of 1 and zero padding is assumed. Let $\mathbf{z}_{j:j+k-1} \in R^k$ denote the vector formed by considering the indices $j$ to $j + k - 1$ of the original vector $\mathbf{z} \in \mathbb{R}^d$. In this case, the convolution operation can be written as

$$\mathbf{W} * \mathbf{x} = \begin{pmatrix} \langle \mathbf{W}_{1\bullet}, \mathbf{x}_{1:k} \rangle & \dots & \langle \mathbf{W}_{1\bullet}, \mathbf{x}_{d-k+1:d} \rangle \\ \vdots & & \vdots \\ \langle \mathbf{W}_{m\bullet}, \mathbf{x}_{1:k} \rangle & \dots & \langle \mathbf{W}_{m\bullet}, \mathbf{x}_{d-k+1:d} \rangle \end{pmatrix} \in \mathbb{R}^{m \times (d-k+1)},$$

where $\mathbf{W}_{i\bullet}$ denotes the $i$-th row of $\mathbf{W}$.

Let us further introduce Toeplitz matrices, $\left\{ \mathbf{T}^{\mathbf{W}_{i\bullet}} \right\}_{i=1}^m$, for each filter with $\mathbf{T}^{\mathbf{W}_{i\bullet}} := \mathrm{toep}(\mathbf{W}_{i\bullet}, d) \in \mathbb{R}^{(d-k+1) \times d}$ such that,

$$\mathbf{T}^{\mathbf{W}_{i\bullet}} = \begin{pmatrix} \mathbf{W}_{i1} & \dots & \mathbf{W}_{ik} & 0 & \dots & 0 \\ 0 & \mathbf{W}_{i1} & \dots & \mathbf{W}_{ik} & 0 & \vdots \\ \vdots & 0 & \ddots & \ddots & \ddots & 0 \\ 0 & \dots & 0 & \mathbf{W}_{i1} & \dots & \mathbf{W}_{ik} \end{pmatrix}.$$

Finally, by stacking the Toeplitz matrices in a row-wise fashion, that is

$$\mathbf{T}^{\mathbf{W}} := \begin{pmatrix} \mathbf{T}^{\mathbf{W}_{1\bullet}} \\ \dots \\ \mathbf{T}^{\mathbf{W}_{m\bullet}} \end{pmatrix} \in \mathbb{R}^{m(d-k+1) \times d},$$

we see that the matrix multiplication of $\mathbf{T}^{\mathbf{W}}$ with an input $\mathbf{x}$ gives the same result as vectorizing the convolution operation row-wise, i.e.

$$\mathrm{vec}_r(\mathbf{W} * \mathbf{x}) = \mathbf{T}^{\mathbf{W}} \mathbf{x}.$$

For a linear network with $L$ hidden layers, each of which is a convolutional kernel, the network function can be formally represented as

$$F_\theta(\mathbf{x}) = \mathcal{W}^{(L+1)} * \mathcal{W}^{(L)} * \dots * \mathcal{W}^{(1)} * \mathbf{x}, \tag{15}$$

where the parameters of each hidden layer $\ell$ are denoted by $\mathcal{W}^{(\ell)} \in \mathbb{R}^{m_\ell \times m_{\ell-1} \times k_l}$. $m_\ell$ is the number of output channels, $m_{\ell-1}$ the number of input channels, and $k_\ell$ the kernel size. As we assumed $\mathbf{x}$ to be one-dimensional, we have $m_0 = 1$.

Similar to a single convolutional operation, the output of a single convolutional layer can be expressed via Toeplitz matrices. The Toeplitz matrix associated with the $\ell$-th convolutional layer $\mathcal{W}^{(\ell)}$ can be expressed by

$$\mathbf{T}^{(\ell)} := \begin{pmatrix} \mathbf{T}^{\mathcal{W}^{(\ell)}_{(1,1)\bullet}} & \dots & \mathbf{T}^{\mathcal{W}^{(\ell)}_{(1,m_{\ell-1})\bullet}} \\ \vdots & & \dots \\ \mathbf{T}^{\mathcal{W}^{(\ell)}_{(m_\ell,1)\bullet}} & \dots & \mathbf{T}^{\mathcal{W}^{(\ell)}_{(m_\ell,m_{\ell-1})\bullet}} \end{pmatrix} \in \mathbb{R}^{m_\ell d_\ell \times m_{\ell-1} d_{\ell-1}}$$

where $\mathcal{W}^{(\ell)}_{(i,j)\bullet} \in \mathbb{R}^{k_\ell}$ refers to the $(i,j)$-th fibre of $\mathcal{W}^{(\ell)}$ and $\mathbf{T}^{\mathcal{W}^{(\ell)}_{(i,j)\bullet}} := \mathrm{toep}(\mathcal{W}^{(\ell)}_{(i,j)\bullet}, d_{\ell-1}) \in \mathbb{R}^{d_\ell \times d_{\ell-1}}$ to itsassociated Toeplitz matrix. Now, the network function can be equivalently written as

$$F_\theta(\mathbf{x}) = \mathbf{T}^{(L+1)} \mathbf{T}^{(L)} \dots \mathbf{T}^{(1)} \mathbf{x}. \tag{16}$$

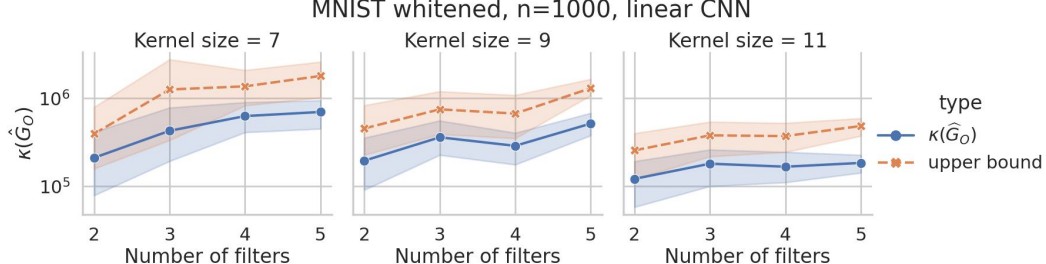

Figure 8: Experiments on the condition number of the Gauss Newton matrix at initialization for a linear two-layer CNN at initialization on a random subsample of n=1000 of MNIST, whitened, with varying kernel size and number of filters. We can see a trend where the number of filters increases the condition number (in analogy to depth in MLPs) and the kernel size improves conditioning (in analogy to width in MLPs).

We further provide experiments on linear CNNs in Figure 8 at initialization showcasing that we can observe similar trends where the number of filters increases the condition number of the Gauss Newton matrix (in analogy to depth in MLPs) and the kernel size improves conditioning (in analogy to width in MLPs).

## C Sensitivity of condition number on choice of smallest eigenvalue

This work considers the pseudo condition number of some matrix $\mathbf{A} \in \mathbb{R}^{n \times n}$, which is the ratio of the largest eigenvalue over the smallest non-zero eigenvalue $\frac{\lambda_{\max}(\mathbf{A})}{\lambda_{\mathrm{nz,min}}(\mathbf{A})}$. If $\mathbf{A}$ is of full rank, the pseudo condition number is equivalent to the condition number. However, if $\mathbf{A}$ is rank deficient, it has to be typically estimated numerically if the rank is not known analytically. Based on the numerical estimation of the matrix rank, the smallest eigenvalue is chosen accordingly and the pseudo condition number is calculated.

In this section we evaluate the sensitivity of choice of the smallest eigenvalue on the resulting pseudo condition number. This is illustrated in the example of a 1-hidden layer network with ReLU activation function and a hidden width of $m = 20$ at initialization and after 100 epochs of training. The details of this experimental setup are further specified in Appendix I.2.
In Figure 9 we can see that depending on the eigenvalue distribution, the condition number can be quite robust (Epoch=0), but also very sensitive (Epoch=100) to the numerical matrix rank estimation.

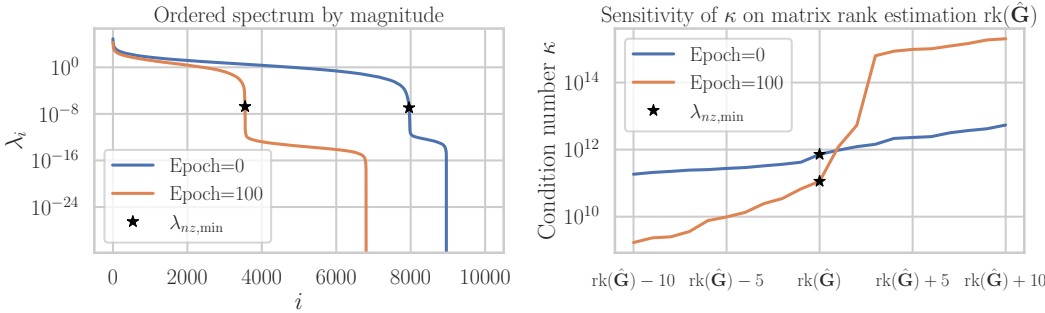

Figure 9: The spectrum of the GN matrix ordered by magnitude at initialization and after 100 epochs. The star marks the smallest non-zero eigenvalue, which is determined by the computed matrix rank (left). Sensitivity of condition number as a function of the matrix rank (right).

# D   Evolution of the condition number during training

We conducted an experiment to track the condition number and evaluate how tight the upper bound is throughout training. For this we trained a 3-layer linear network with a hidden width of $m = 500$ with three different seeds for 5000 epochs with SGD with a mini-batch size of 256 and a constant learning rate of 0.2 on a subset of Cifar-10 ($n = 1000$) [Krizhevsky et al., 2009], which has been downsampled to $3 \times 8 \times 8$ images and whitened. The network was trained on a single NVIDIA GeForce RTX 3090 GPU and took around 5 minutes per run. The condition number was computed via the explicit formula in Lemma 2 on CPU and took around 4 minutes. The condition number and the upper bound are shown in Figure 10 together with the training loss. We make the following two observations: 1. The condition number takes values in the range of 6 to 12 throughout training, which corresponds to a maximal change of around 50% of the condition number at initialization. This indicates that the condition number at initialization can be indicative of how the conditioning is along the optimization trajectory. 2. The upper bound remains tight throughout training, which highlights that it can provide reliable information about the condition number of the Gauss-Newton matrix throughout training.

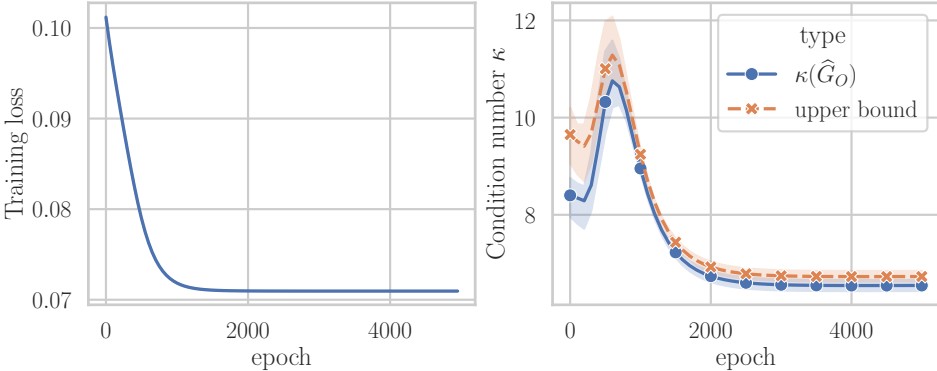

Figure 10: Training loss (left) and corresponding evolution of the condition number throughout training for three seeds. The shaded area in the figures corresponds to one standard deviation from the mean.

## E Understanding the difference between the convex combination bound and the maximum Bound

In Lemma 2 we have seen that the upper bound of the condition number can be expressed in terms of a convex combination of the condition number of the weight matrices. In this section, we will elucidate the 'self-balancing' behavior of the individual terms in Lemma 2. For completeness, let us restate Lemma 2 here again:

**Lemma 2.** *Assume that $m > \max\{d, k\}$ and that $\alpha_\ell := \sigma^2_{\min}(\mathbf{W}^{L:\ell+1}) \cdot \sigma^2_{\min}(\mathbf{W}^{1:\ell-1}) > 0 \ \forall \ell = 1, \ldots, L$. Let $\gamma_\ell := \frac{\alpha_\ell}{\sum_{i=1}^{L} \alpha_i}$. Then the condition number of the GN matrix of a $L$-layer linear network can be upper-bounded in the following way:*

$$\kappa(\widehat{\mathbf{G}}_O) \leq \kappa(\mathbf{\Sigma}) \sum_{\ell=1}^{L} \gamma_\ell \kappa(\mathbf{W}^{L:\ell+1})^2 \kappa(\mathbf{W}^{1:\ell-1})^2 \leq \kappa(\mathbf{\Sigma}) \max_{1 \leq \ell \leq L} \left\{ \kappa(\mathbf{W}^{L:\ell+1})^2 \kappa(\mathbf{W}^{1:\ell-1})^2 \right\}.$$

By looking at the individual values of $\kappa^2(\mathbf{W}^{L:\ell+1})$ and $\sigma^2_{\min}(\mathbf{W}^{L:\ell+1})$ and similarly $\kappa^2(\mathbf{W}^{1:\ell-1})$ and $\sigma^2_{\min}(\mathbf{W}^{1:\ell-1})$ over $\ell = 1, \ldots, L$ for networks of increasing depth $L$, we can see the 'self-balancing' behavior in Figure 11 for downsampled MNIST ($d = 196$) over 10 runs and $m = 300$ neurons per hidden layer. Note that this behavior is also consistent for different widths $m$, input dimension $d$, and output dimension $k$. By first looking at the first row, we observe that $\kappa^2(\mathbf{W}^{1:\ell-1})$ (top left) dominates the product of each term in Lemma 2 and follows an exponential growth rule in $\ell$ (note the log scale of the $y$-axis). By looking at the second row, we see that at the same time, the smallest singular value of $\kappa^2(\mathbf{W}^{1:\ell-1})$ (bottom left) is decreasing exponentially, leading to the 'self-balancing' behavior of the condition number. The same observation also holds for $\mathbf{W}^{L:\ell+1}$, but since the number of weight matrices decreases with $\ell$ this leads to a mirrored plot. The 'self-balancing' behavior can be seen below in Figure 12 for ease of reading, where the individual terms of Lemma 2 are plotted with and without the weighting factor $\gamma_\ell$.

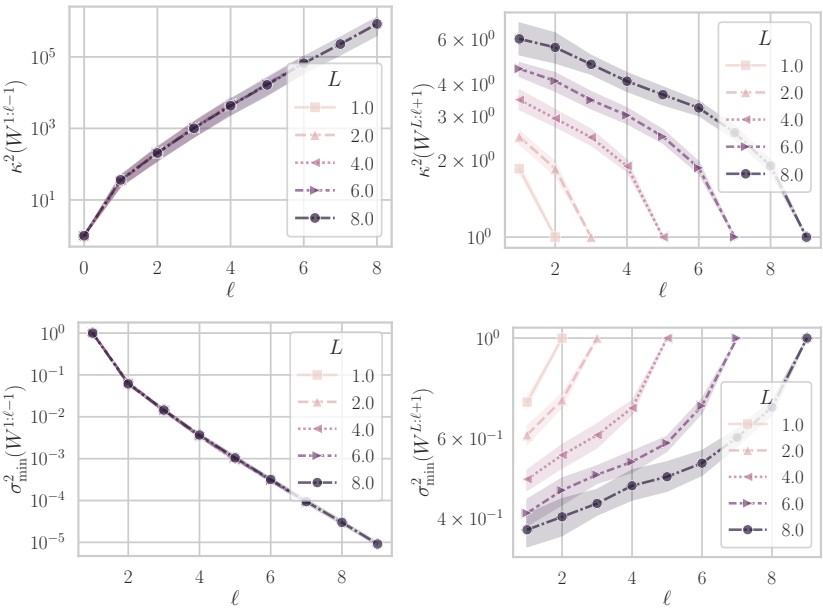

Figure 11: Condition number and smallest eigenvalue of $\mathbf{W}^{1:\ell-1}$ (first column) and $\mathbf{W}^{L:\ell+1}$ (second column) for downsampled MNIST ($d = 196$) for three seeds. Shaded area corresponds to one standard deviation. Note that the y-axis is log-scaled at the different limits for each subplot.

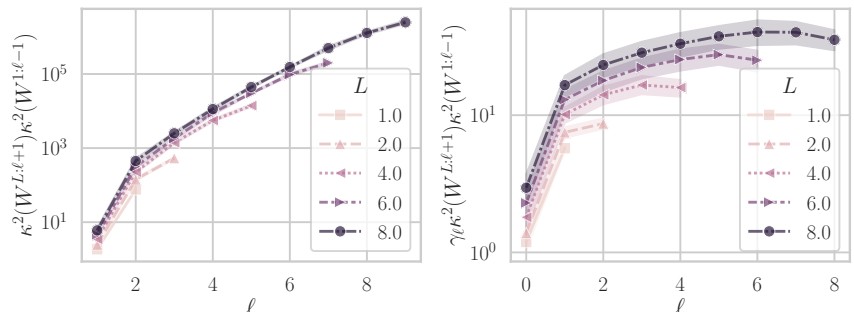

Figure 12: Condition number of weight matrices with and without weighting factor $\gamma_\ell$ for downsampled MNIST ($d = 196$) for three seeds, which highlights the 'self-balancing' effect of the weight condition number. Note that the y-axis is log-scaled and both plots have different limits. Shaded area corresponds to one standard deviation.

## F  Conditioning of functional Hessian for one-hidden layer network

Consider the case of a one-hidden layer linear network $F_{\boldsymbol{\theta}}(\mathbf{x}) = \mathbf{W}\mathbf{V}\mathbf{x}$, with $\mathbf{x} \in \mathbb{R}^d, \mathbf{W} \in \mathbb{R}^{k \times m}$, and $\mathbf{V} \in \mathbb{R}^{m \times d}$ and some corresponding label $\mathbf{y} \in \mathbb{R}^k$. Under the assumption of the MSE loss, the functional Hessian is then given as:

$$\mathbf{H}_{\mathrm{F}} = \begin{pmatrix} \mathbf{0}_{km \times km} & \boldsymbol{\Omega} \otimes \mathbf{I}_m \\ \boldsymbol{\Omega}^\top \otimes \mathbf{I}_m & \mathbf{0}_{dm \times dm,} \end{pmatrix} \tag{17}$$

where $\boldsymbol{\Omega} = \mathbb{E}[\delta_{\mathbf{x},\mathbf{y}}\mathbf{x}^\top] \in \mathbb{R}^{k \times d}$ is the residual-input covariance matrix, with $\delta_{\mathbf{x},\mathbf{y}} := F_{\boldsymbol{\theta}}(\mathbf{x}) - \mathbf{y}$ being the residual. Since we would like to estimate the spectrum of $\mathbf{H}_{\mathrm{F}}$, let us suppose that it has an eigenvector $\mathbf{v} = \begin{pmatrix} \mathbf{a} \otimes \mathbf{b} \\ \mathbf{c} \otimes \mathbf{d} \end{pmatrix}$. Assuming that $\lambda$ is the corresponding eigenvalue, we have the following eigenproblem:

$$\mathbf{H}_{\mathrm{F}} \cdot \mathbf{v} = \begin{pmatrix} \boldsymbol{\Omega}\,\mathbf{c} \otimes \mathbf{d} \\ \boldsymbol{\Omega}^\top \mathbf{a} \otimes \mathbf{b} \end{pmatrix} = \lambda \begin{pmatrix} \mathbf{a} \otimes \mathbf{b} \\ \mathbf{c} \otimes \mathbf{d} \end{pmatrix}$$

Let's start with the guess that $\mathbf{b} = \mathbf{d}$, then we obtain the following equations:

$$\lambda\,\mathbf{a} = \boldsymbol{\Omega}\,\mathbf{c} \quad \text{and} \quad \lambda\,\mathbf{c} = \boldsymbol{\Omega}^\top \mathbf{a}.$$

Solving this gives:

$$\boldsymbol{\Omega}^\top \boldsymbol{\Omega}\,\mathbf{c} = \lambda^2\,\mathbf{c} \quad \text{and} \quad \boldsymbol{\Omega}\boldsymbol{\Omega}^\top \mathbf{a} = \lambda^2\,\mathbf{a}.$$

Hence, $\mathbf{a}$ and $\mathbf{c}$ are simply the eigenvectors of $\boldsymbol{\Omega}\boldsymbol{\Omega}^\top$ and $\boldsymbol{\Omega}^\top \boldsymbol{\Omega}$ respectively. Both these matrices have the same non-zero eigenvalues, and the resulting eigenvalues for the functional Hessian are nothing but the square root of corresponding eigenvalues of the matrix $\boldsymbol{\Omega}^\top \boldsymbol{\Omega}$ — with both positive and negative signs. About $\mathbf{b} = \mathbf{d}$: We still have a degree of freedom left in choosing them. Any $m$-orthogonal vectors with unit norm would satisfy the above equation and, for simplicity, we can just pick the canonical basis vectors in $\mathbb{R}^m$.

**Teacher-Student setting**  Let us now further assume a Teacher-student setting, i.e. $\mathbf{y} = \mathbf{Z}\mathbf{x}$ for some $\mathbf{Z} \in \mathbb{R}^{k \times d}$. In this case note that $\boldsymbol{\Omega} = \mathbb{E}[(\mathbf{W}\mathbf{V}\mathbf{x} - \mathbf{y})\mathbf{x}^\top] = (\mathbf{W}\mathbf{V} - \mathbf{Z})\mathbb{E}[\mathbf{x}\mathbf{x}^\top] = (\mathbf{W}\mathbf{V} - \mathbf{Z})\boldsymbol{\Sigma}$. Now recall the definition of the condition number $\kappa(\mathbf{H}_F) = \|\mathbf{H}_F\| \cdot \|\mathbf{H}_F^{-1}\|$ and using the observation

from above we have that

$$\|\mathbf{H}_F\| = \sigma_{\max}(\mathbf{H}_F) = \sigma_{\max}((\mathbf{WV} - \mathbf{Z})\mathbf{\Sigma})$$
$$\leq \sigma_{\max}(\mathbf{WV} - \mathbf{Z}) \cdot \lambda_{\max}(\mathbf{\Sigma})$$
$$\|\mathbf{H}_F^{-1}\| = \sigma_{\min}(\mathbf{H}_F) = \sigma_{\min}((\mathbf{WV} - \mathbf{Z})\mathbf{\Sigma})$$
$$\geq \sigma_{\min}(\mathbf{WV} - \mathbf{Z}) \cdot \lambda_{\min}(\mathbf{\Sigma}).$$

Therefore we have

$$\kappa(\mathbf{H}_F) \leq \kappa(\mathbf{WV} - \mathbf{Z}) \cdot \kappa(\mathbf{\Sigma}). \tag{18}$$

Similar to bounding the condition number of the GN matrix, we can also clearly see a dependency on the conditioning of the input covariance.

## G  Concrete bounds at initialization for one-hidden layer linear network

### G.1  Non-asymptotic bound

Based on the upper bound in Eq. (6) derived for the one-hidden layer linear network, we can look at how this bound behaves concretely at initialization. If we assume the entries of $\mathbf{W}$ and $\mathbf{V}$ to be i.i.d. Gaussian entries, then $\mathbf{WW}^\top$ and $\mathbf{V}^\top\mathbf{V}$ will be Wishart matrices, for which asymptotically the distribution of the smallest and largest eigenvalues are known. Also there are known non-asymptotic bounds to bound the maximum and minimum singular value of a Gaussian matrix:

**Lemma 6** (Corollary 5.35 in Vershynin [2010]). *Let $\mathbf{A}$ be an $N \times n$ matrix whose entries are independent standard normal random variables $\mathcal{N}(0,1)$. Then for every $t \geq 0$, with probability at least $1 - 2\exp(-\frac{t^2}{2})$ one has*

$$\sqrt{N} - \sqrt{n} - t \leq \sigma_{\min}(\mathbf{A}) \leq \sigma_{\max}(\mathbf{A}) \leq \sqrt{N} + \sqrt{n} + t. \tag{19}$$

Using the Lemma above and a union bound, we can also derive a non-asymptotic bound for the condition number of the GN matrix under the assumption that we have a wide layer $m > \max(d, k)$:

**Lemma 7.** *Assume that $m \geq \max(d, k)$ and that the entries of $\mathbf{V} \in \mathbb{R}^{m \times d}, \mathbf{W} \in \mathbb{R}^{k \times m}$ are initialized Gaussian i.i.d. with $\mathbf{V}_{ij} \sim \mathcal{N}(0, \sigma_v^2)$ and $\mathbf{W}_{ij} \sim \mathcal{N}(0, \sigma_w^2)$, then with probability at least $1 - 8\exp(-\frac{t^2}{2})$ the condition number of the GN matrix $\kappa(\widehat{\mathbf{G}}_O)$ is upper bounded by*

$$\kappa(\widehat{\mathbf{G}}_O) = \frac{\lambda_{\max}(\widehat{\mathbf{G}}_O)}{\lambda_{\min}(\widehat{\mathbf{G}}_O)} \leq \kappa(\mathbf{\Sigma}) \frac{(\sigma_w^2(\sqrt{m} + \sqrt{k} + t)^2 + \sigma_v^2(\sqrt{m} + \sqrt{d} + t)^2)}{(\sigma_w^2(\sqrt{m} - \sqrt{k} - t)^2 + \sigma_v^2(\sqrt{m} - \sqrt{d} - t)^2)}.$$

*Proof.* Assume that $\mathbf{V} \in \mathbb{R}^{m \times d}, \mathbf{W} \in \mathbb{R}^{k \times m}$ are initialized Gaussian i.i.d. with $\mathbf{V}_{ij} \sim \mathcal{N}(0, \sigma_v^2)$ and $\mathbf{W}_{ij} \sim \mathcal{N}(0, \sigma_w^2)$, then the bounds in Lemma 6 are scaled with $\sigma_v$ and $\sigma_w$ respectively:

$$\sigma_v(\sqrt{m} - \sqrt{d} - t) \leq \sigma_{\min}(\mathbf{V}) \leq \sigma_{\max}(\mathbf{V}) \leq \sigma_v(\sqrt{m} + \sqrt{d} + t)$$
$$\sigma_w(\sqrt{m} - \sqrt{k} - t) \leq \sigma_{\min}(\mathbf{W}) \leq \sigma_{\max}(\mathbf{W}) \leq \sigma_w(\sqrt{m} + \sqrt{k} + t).$$

Further noting that

$$\lambda_{\min}(\mathbf{V}^\top\mathbf{V}) = \sigma_{\min}^2(\mathbf{V}), \quad \lambda_{\max}(\mathbf{V}^\top\mathbf{V}) = \sigma_{\max}^2(\mathbf{V})$$
$$\lambda_{\min}(\mathbf{WW}^\top) = \sigma_{\min}^2(\mathbf{W}) \quad \lambda_{\max}(\mathbf{WW}^\top) = \sigma_{\max}^2(\mathbf{W}),$$

we get the following non-asymptotic upper respectively lower bounds for the extreme eigenvalues with probability at least $1 - 2\exp(-\frac{t^2}{2})$ each:

$$\lambda_{\min}(\mathbf{V}^\top\mathbf{V}) \geq \sigma_v^2(\sqrt{m} - \sqrt{d} - t)^2, \quad \lambda_{\max}(\mathbf{V}^\top\mathbf{V}) \leq \sigma_v^2(\sqrt{m} + \sqrt{d} + t)^2$$
$$\lambda_{\min}(\mathbf{WW}^\top) \geq \sigma_w^2(\sqrt{m} - \sqrt{k} - t)^2, \quad \lambda_{\max}(\mathbf{WW}^\top) \leq \sigma_w^2(\sqrt{m} + \sqrt{k} + t)^2$$

Using a union bound, we get the upper bound for the condition number of the GN matrix. $\qquad\square$

Based on the bound derived above, we can understand how the condition number behaves under initialization schemes used in practice. Consider the setting of a wide hidden layer $m \gg \max\{d, k\}$, in which the input dimension is much larger than the output dimension, i.e. $d \gg k$. This is for instance the case for image classification, where the number of image pixels is typically much larger than the number of image classes. A popular initialization scheme is the one proposed in He et al. [2015], commonly known as the Kaiming/He initialization, which corresponds to $\sigma_w^2 = \frac{1}{m}, \sigma_v^2 = \frac{1}{d}$ in our case. Hence, under the assumption of $m \gg d$, we will have $\sigma_v^2 \gg \sigma_w^2$ and therefore

$$\kappa(\widehat{\mathbf{G}}_O) \leq \kappa(\mathbf{\Sigma}) \frac{(\sigma_w^2(\sqrt{m} + \sqrt{k} + t)^2 + \sigma_v^2(\sqrt{m} + \sqrt{d} + t)^2)}{(\sigma_w^2(\sqrt{m} - \sqrt{k} - t)^2 + \sigma_v^2(\sqrt{m} - \sqrt{d} - t)^2)} \tag{20}$$

$$\approx \kappa(\mathbf{\Sigma}) \frac{(\sqrt{m} + \sqrt{d} + t)^2}{(\sqrt{m} - \sqrt{d} - t)^2} \approx \kappa(\mathbf{\Sigma}). \tag{21}$$

## G.2 Asymptotic bound

To complement the analysis, we can also derive an asymptotic bound for the condition number when $d, k, m$ go to infinity together.

For this, we will use a result from Silverstein [1985], in which it is shown that the smallest eigenvalue of the matrix $\mathbf{M}_s = \frac{1}{s}\mathbf{U}_s\mathbf{U}_s^\top$, where $\mathbf{U}_s \in \mathbb{R}^{n \times s}$ composed of i.i.d. $\mathcal{N}(0, 1)$ random variables, converges to $(1 - \sqrt{y})^2$ as $s \to \infty$, while $\frac{n}{s} \to y \in (0, 1)$ as $s \to \infty$. This corresponds to the case in which the hidden layer is larger than the input and output dimensions, that is $m \geq \max(d, k)$. Similarly it can be shown that $\lambda_{\max} \to (1 + \sqrt{y})^2$ for all $y > 0$ for $s \to \infty$.

Using this result we can bound the extreme eigenvalues of $\mathbf{W}\mathbf{W}^\top$ and $\mathbf{V}^\top\mathbf{V}$ at initialization. Assuming that the entries of $\mathbf{W}$ and $\mathbf{V}$ are entry-wise independent normally distributed $\mathbf{W}_{ij} \sim \mathcal{N}(0, \sigma_w^2), \mathbf{V}_{ij} \sim \mathcal{N}(0, \sigma_v^2)$ (for instance $\sigma_w^2 = \frac{1}{m}, \sigma_v^2 = \frac{1}{d}$ for Kaiming-initialization) we get for $d, k, m \to \infty$:

$$\lambda_{\min}(\mathbf{W}\mathbf{W}^\top) = \sigma_w^2 \cdot m \cdot \lambda_{\min}\left(\frac{1}{m}\mathbf{U}_m\mathbf{U}_m^\top\right) \overset{k,m\to\infty}{\to} \sigma_w^2 \cdot m \cdot \left(1 - \sqrt{\frac{k}{m}}\right)^2$$

$$\lambda_{\max}(\mathbf{W}\mathbf{W}^\top) = \sigma_w^2 \cdot m \cdot \lambda_{\min}\left(\frac{1}{m}\mathbf{U}_m\mathbf{U}_m^\top\right) \overset{k,m\to\infty}{\to} \sigma_w^2 \cdot m \cdot \left(1 + \sqrt{\frac{k}{m}}\right)^2$$

$$\lambda_{\min}(\mathbf{V}\mathbf{V}^\top) = \sigma_v^2 \cdot m \cdot \lambda_{\min}\left(\frac{1}{m}\mathbf{U}_m\mathbf{U}_m^\top\right) \overset{d,m\to\infty}{\to} \sigma_v^2 \cdot m \cdot \left(1 - \sqrt{\frac{d}{m}}\right)^2$$

$$\lambda_{\max}(\mathbf{V}\mathbf{V}^\top) = \sigma_v^2 \cdot m \cdot \lambda_{\max}\left(\frac{1}{m}\mathbf{U}_m\mathbf{U}_m^\top\right) \overset{d,m\to\infty}{\to} \sigma_v^2 \cdot m \cdot \left(1 + \sqrt{\frac{d}{m}}\right)^2.$$

Thus, in the asymptotic case in which the hidden layer is wide $m \geq \max\{d, k\}$, we can derive the following upper bound for the condition number:

$$\kappa(\widehat{\mathbf{G}}_O) \leq C \overset{d,k,m\to\infty}{\to} \kappa(\mathbf{\Sigma}) \cdot \frac{\sigma_w^2 \cdot \left(1 + \sqrt{\frac{k}{m}}\right)^2 + \sigma_v^2 \cdot \left(1 + \sqrt{\frac{d}{m}}\right)^2}{\sigma_w^2 \cdot \left(1 - \sqrt{\frac{k}{m}}\right)^2 + \sigma_v^2 \cdot \left(1 - \sqrt{\frac{d}{m}}\right)^2} \tag{22}$$

Again, if $m \gg \max\{d, k\}$ we also have in the asymptotic limit that $\kappa(\widehat{\mathbf{G}}_O) \approx \kappa(\mathbf{\Sigma})$.

## H   Proofs

**Lemma 1.** *Let $\beta_w = \sigma_{\min}^2(\mathbf{W})/(\sigma_{\min}^2(\mathbf{W}) + \sigma_{\min}^2(\mathbf{V}))$. Then the condition number of GN for the one-hidden layer network with linear activations with $m > \max\{d, k\}$ is upper bounded by*

$$\kappa(\widehat{\mathbf{G}}_O) \leq \kappa(\mathbf{\Sigma}) \cdot \frac{\sigma_{\max}^2(\mathbf{W}) + \sigma_{\max}^2(\mathbf{V})}{\sigma_{\min}^2(\mathbf{W}) + \sigma_{\min}^2(\mathbf{V})} = \kappa(\mathbf{\Sigma}) \cdot (\beta_w \, \kappa(\mathbf{W})^2 + (1 - \beta_w) \, \kappa(\mathbf{V})^2). \quad (6)$$

*Proof.* Note that we have by the Weyl's in dual Weyl's inequality that

$$\lambda_{\max}(\widehat{\mathbf{G}}_O) = \lambda_{\max}(\mathbf{W}\mathbf{W}^\top \otimes \mathbf{\Sigma} + \mathbf{I}_k \otimes \mathbf{\Sigma}^{1/2}\mathbf{V}^\top\mathbf{V}\mathbf{\Sigma}^{1/2}) \leq \lambda_{\max}(\mathbf{W}\mathbf{W}^\top \otimes \mathbf{\Sigma}) + \lambda_{\max}(\mathbf{I}_k \otimes \mathbf{\Sigma}^{1/2}\mathbf{V}^\top\mathbf{V}\mathbf{\Sigma}^{1/2})$$

$$\lambda_{\min}(\widehat{\mathbf{G}}_O) = \lambda_{\min}(\mathbf{W}\mathbf{W}^\top \otimes \mathbf{\Sigma} + \mathbf{I}_k \otimes \mathbf{\Sigma}^{1/2}\mathbf{V}^\top\mathbf{V}\mathbf{\Sigma}^{1/2}) \geq \lambda_{\min}(\mathbf{W}\mathbf{W}^\top \otimes \mathbf{\Sigma}) + \lambda_{\min}(\mathbf{I}_k \otimes \mathbf{\Sigma}^{1/2}\mathbf{V}^\top\mathbf{V}\mathbf{\Sigma}^{1/2}).$$

Then note that $\mathbf{W}\mathbf{W}^\top \otimes \mathbf{\Sigma}$ and $\mathbf{I}_k \otimes \mathbf{\Sigma}^{1/2}\mathbf{V}^\top\mathbf{V}\mathbf{\Sigma}^{1/2}$ are positive semidefinite and using the equality for the extreme eigenvalues of Kronecker products in Eq. (13), we have the bound

$$\lambda_{\max}(\mathbf{W}\mathbf{W}^\top \otimes \mathbf{\Sigma}) + \lambda_{\max}(\mathbf{I}_k \otimes \mathbf{\Sigma}^{1/2}\mathbf{V}^\top\mathbf{V}\mathbf{\Sigma}^{1/2}) = \lambda_{\max}(\mathbf{W}\mathbf{W}^\top)\lambda_{\max}(\mathbf{\Sigma}) + \lambda_{\max}(\mathbf{\Sigma}^{1/2}\mathbf{V}^\top\mathbf{V}\mathbf{\Sigma}^{1/2})$$

$$\lambda_{\min}(\mathbf{W}\mathbf{W}^\top \otimes \mathbf{\Sigma}) + \lambda_{\min}(\mathbf{I}_k \otimes \mathbf{\Sigma}^{1/2}\mathbf{V}^\top\mathbf{V}\mathbf{\Sigma}^{1/2}) = \lambda_{\min}(\mathbf{W}\mathbf{W}^\top)\lambda_{\min}(\mathbf{\Sigma}) + \lambda_{\min}(\mathbf{\Sigma}^{1/2}\mathbf{V}^\top\mathbf{V}\mathbf{\Sigma}^{1/2}).$$

Using the submultiplicativity of the matrix norm we get the upper bound

$$\kappa(\widehat{\mathbf{G}}_O) \leq \kappa(\mathbf{\Sigma}) \cdot \frac{\lambda_{\max}(\mathbf{W}\mathbf{W}^\top) + \lambda_{\max}(\mathbf{V}^\top\mathbf{V})}{\lambda_{\min}(\mathbf{W}\mathbf{W}^\top) + \lambda_{\min}(\mathbf{V}^\top\mathbf{V})}.$$

Finally, by noting that we have $\lambda_i(\mathbf{W}\mathbf{W}^\top) = \sigma_i^2(\mathbf{W})$ and $\lambda_i(\mathbf{V}^\top\mathbf{V}) = \sigma_i^2(\mathbf{V})$ for $i \in \{\min, \max\}$ because we have $m > \max\{d, k\}$ yields the result. $\qquad\square$

**Lemma 4.** *Consider an one-hidden layer network with a Leaky-ReLU activation with negative slope $\alpha \in [0, 1]$. Then the condition number of the GN matrix defined in Equation (10) can be bounded as:*

$$\kappa(\widehat{\mathbf{G}}_O) \leq \frac{\sigma_{\max}^2(\mathbf{X}) \, \sigma_{\max}^2(\mathbf{W}) + \lambda_{\max}(\mathbf{\Gamma})}{\alpha^2 \, \sigma_{\min}^2(\mathbf{X}) \, \sigma_{\min}^2(\mathbf{W}) + \lambda_{\min}(\mathbf{\Gamma})} \quad (11)$$

*Proof.* As before we will bound the condition number by separately bounding the extreme eigenvalues first.

First, recall the expression of the GN matrix as

$$\widehat{\mathbf{G}}_O = \sum_{i=1}^m \mathbf{\Lambda}^i\mathbf{X}^\top\mathbf{X}\mathbf{\Lambda}^i \otimes \mathbf{W}_{\bullet i}\mathbf{W}_{\bullet i}^\top + \sum_{i=1}^m \mathbf{\Lambda}^i\mathbf{X}^\top\mathbf{V}_{i\bullet}^\top\mathbf{V}_{i\bullet}\mathbf{X}\mathbf{\Lambda}^i \otimes \mathbf{I}_k$$

and note that we can apply the Weyl's and dual Weyl's inequality because $(\mathbf{\Lambda}^i\mathbf{X}^\top\mathbf{X}\mathbf{\Lambda}^i) \otimes (\mathbf{W}_{\bullet,i}\mathbf{W}_{\bullet,i}^\top)$ and $(\mathbf{\Lambda}^i\mathbf{X}^\top\mathbf{V}_{i,\bullet}\mathbf{V}_{i,\bullet}^\top\mathbf{X}\mathbf{\Lambda}^i) \otimes \mathbf{I}_k$ are symmetric, yielding the following bounds:

$$\lambda_{\min}(\widehat{\mathbf{G}}_O) \geq \lambda_{\min}\left(\sum_{i=1}^m (\mathbf{\Lambda}^i\mathbf{X}^\top\mathbf{X}\mathbf{\Lambda}^i) \otimes (\mathbf{W}_{\bullet,i}\mathbf{W}_{\bullet,i}^\top)\right) + \lambda_{\min}\left(\sum_{i=1}^m (\mathbf{\Lambda}^i\mathbf{X}^\top\mathbf{V}_{i,\bullet}^\top\mathbf{V}_{i,\bullet}\mathbf{X}\mathbf{\Lambda}^i) \otimes \mathbf{I}_k\right)$$

$$\lambda_{\max}(\widehat{\mathbf{G}}_O) \leq \lambda_{\max}\left(\sum_{i=1}^m (\mathbf{\Lambda}^i\mathbf{X}^\top\mathbf{X}\mathbf{\Lambda}^i) \otimes (\mathbf{W}_{\bullet,i}\mathbf{W}_{\bullet,i}^\top)\right) + \lambda_{\max}\left(\sum_{i=1}^m (\mathbf{\Lambda}^i\mathbf{X}^\top\mathbf{V}_{i,\bullet}^\top\mathbf{V}_{i,\bullet}\mathbf{X}\mathbf{\Lambda}^i) \otimes \mathbf{I}_k\right).$$

Then we can bound the first term by using the fact that for the sum of a Kronecker product of PSD matrices $\mathbf{A}_i, \mathbf{B}_i$ for $i = 1, \ldots, m$ we have $\lambda_{\min}\left(\sum_{i=1}^m (\mathbf{A}_i \otimes \mathbf{B}_i)\right) \geq \min_{1 \leq j \leq m} \lambda_{\min}(\mathbf{A}_j) \cdot \lambda_{\min}\left(\sum_{i=1}^m \mathbf{B}_i\right)$ to get

$$\lambda_{\min}\left(\sum_{i=1}^m (\mathbf{\Lambda}^i\mathbf{X}^\top\mathbf{X}\mathbf{\Lambda}^i) \otimes (\mathbf{W}_{\bullet,i}\mathbf{W}_{\bullet,i}^\top)\right) \geq \lambda_{\min}(\mathbf{\Lambda}^i\mathbf{X}^\top\mathbf{X}\mathbf{\Lambda}^i)\lambda_{\min}\left(\sum_{i=1}^m \mathbf{W}_{\bullet,i}\mathbf{W}_{\bullet,i}^\top\right)$$

$$\geq \alpha^2 \lambda_{\min}(\mathbf{X}^\top\mathbf{X})\lambda_{\min}(\mathbf{W}\mathbf{W}^\top) \quad (23)$$

$$= \alpha^2 \sigma_{\min}^2(\mathbf{X})\sigma_{\min}^2(\mathbf{W}) \quad (24)$$

by noting that $\mathbf{\Lambda}^i$ only contains 1 or $-\alpha$ on its diagonals in case of Leaky-ReLU activation and $\sum_{i=1}^m \mathbf{W}_{\bullet,i}\mathbf{W}_{\bullet,i}^\top = \mathbf{W}\mathbf{W}^\top$. Analogously, we have

$$\lambda_{\max}\left(\sum_{i=1}^m (\mathbf{\Lambda}^i \mathbf{X}^\top \mathbf{X}\mathbf{\Lambda}^i) \otimes (\mathbf{W}_{\bullet,i}\mathbf{W}_{\bullet,i}^\top)\right) \le \sigma_{\max}^2(\mathbf{X})\sigma_{\max}^2(\mathbf{W}).$$

This yields the first upper bound.

To get the second upper bound, we need to further bound the second term. For this, note that

$$\lambda_{\min}\left(\sum_{i=1}^m (\mathbf{\Lambda}^i \mathbf{X}^\top \mathbf{V}_{i,\bullet}^\top \mathbf{V}_{i,\bullet}\mathbf{X}\mathbf{\Lambda}^i) \otimes \mathbf{I}_k\right) = \lambda_{\min}\left(\sum_{i=1}^m (\mathbf{\Lambda}^i \mathbf{X}^\top \mathbf{V}_{i,\bullet}^\top \mathbf{V}_{i,\bullet}\mathbf{X}\mathbf{\Lambda}^i)\right)$$

$$\ge \alpha^2 \lambda_{\min}\left(\sum_{i=1}^m (\mathbf{X}^\top \mathbf{V}_{i,\bullet}^\top \mathbf{V}_{i,\bullet}\mathbf{X})\right)$$

$$= \alpha^2 \lambda_{\min}\left(\mathbf{X}^\top \sum_{i=1}^m (\mathbf{V}_{i,\bullet}^\top \mathbf{V}_{i,\bullet})\mathbf{X}\right)$$

$$= \alpha^2 \lambda_{\min}\left(\mathbf{X}^\top \mathbf{V}^\top \mathbf{V}\mathbf{X}\right). \tag{25}$$

Analogously we have

$$\lambda_{\max}\left(\sum_{i=1}^m (\mathbf{\Lambda}^i \mathbf{X}^\top \mathbf{V}_{i,\bullet}^\top \mathbf{V}_{i,\bullet}\mathbf{X}\mathbf{\Lambda}^i) \otimes \mathbf{I}_k\right) \le \lambda_{\max}\left(\mathbf{X}^\top \mathbf{V}^\top \mathbf{V}\mathbf{X}\right), \tag{26}$$

which gives the second upper bound. $\qquad\square$

### H.1 Further details on the effect of condition number at initialization on convergence rate

We present here further details of the modified analysis of GD for strongly convex functions to study the effect of the condition number at initialization on the convergence rate, where we use local constants $\mu(k)$ and $L(k)$ instead of the global smoothness and Lipschitz constant, respectively. Let $L$ denote the Lipschitz constant and let the smoothness constant be denoted by $\mu$. Furthermore, let the step size be such that $\eta_k \le \frac{1}{L}$. Then by the definition of gradient descent we get

$$||\boldsymbol{\theta}_{k+1} - \boldsymbol{\theta}^*||^2 = ||\boldsymbol{\theta}_k - \boldsymbol{\theta}^* - \eta_k \nabla f(\boldsymbol{\theta}_k)||^2$$
$$= ||\boldsymbol{\theta}_k - \boldsymbol{\theta}^*||^2 - 2\eta_k \nabla f\left(\boldsymbol{\theta}_k\right)^\top (\boldsymbol{\theta}_k - \boldsymbol{\theta}^*) + \eta_k^2 ||\nabla f\left(\boldsymbol{\theta}_k\right)||^2$$
$$\overset{\text{Strong convexity}}{\le} (1 - \eta_k\mu)||\boldsymbol{\theta}_k - \boldsymbol{\theta}^*||^2 - 2\eta_k(f(\boldsymbol{\theta}_k) - f(\boldsymbol{\theta}^*)) + \eta_k^2 ||\nabla f\left(\boldsymbol{\theta}_k\right)||^2$$
$$\overset{\text{Smoothness}}{\le} (1 - \eta_k\mu)||\boldsymbol{\theta}_k - \boldsymbol{\theta}^*||^2 - 2\eta_k(f(\boldsymbol{\theta}_k) - f(\boldsymbol{\theta}^*)) + 2\eta_k^2 L(f(\boldsymbol{\theta}_k) - f(\boldsymbol{\theta}^*))$$
$$= (1 - \eta_k\mu)||\boldsymbol{\theta}_k - \boldsymbol{\theta}^*||^2 - 2\eta_k(1 - \eta_k L)(f(\boldsymbol{\theta}_k) - f(\boldsymbol{\theta}^*))$$

Since we assumed that $\eta_k \le \frac{1}{L}$, the last term is negative. Therefore:

$$||\boldsymbol{\theta}_{k+1} - \boldsymbol{\theta}^*||^2 \le (1 - \eta_k\mu)||\boldsymbol{\theta}_k - \boldsymbol{\theta}^*||^2 \tag{27}$$

So by recursively applying (27) and replacing $\mu$ by the local smoothness constants $\mu(k)$:

$$||\boldsymbol{\theta}_k - \boldsymbol{\theta}^*||^2 \le \prod_{i=0}^{k-1}(1 - \eta_i\mu(i))||\boldsymbol{\theta}_0 - \boldsymbol{\theta}^*||^2. \tag{28}$$

One can see the effect of $\mu(0)$ in the bound, which is even more dominant when $\mu(k)$ changes slowly. Of course, the effect of $\mu(0)$ attenuates over time, and that's why we are talking about a local effect.

# I  Further experimental results

## I.1  Further experiments on pruning networks at initialization

We repeated the experiments of pruning the weights at initialization for different architectures, including a small Vision Transformer (ViT)[3](Figure 13), a ResNet20 (Figure 14), a ResNet32 (Figure 15), a VGG5Net[4] (Figure 16) and a Feed-forward network (Figure 17). The default initialization from PyTorch [Paszke et al., 2019] was used. In all setups, the weights were pruned layer-wise by magnitude and trained on a subset of Cifar-10 of $n = 1000$ images. The ViT was trained with AdamW with a learning rate of $1e^{-2}$ and weight decay $1e^{-2}$. The ResNet20, ResNet32 and VGGnet were trained with SGD with momentum = 0.9 and weight decay of $10^{-4}$ and a learning rate of 0.1 with a step decay to 0.01 after 91 epochs for ResNet20 and ResNet32 and after 100 epochs for the VGGnet. The Feed-forward network was trained with SGD with a constant learning rate of 0.01. All networks were trained with a mini-batch size of 64, the ResNet20, ResNet32 and the Feed-forward network were trained on a single NVIDIA GeForce RTX 3090 GPU and the ViT on a single GPU NVIDIA GeForce RTX 4090. The training time was around 1 hour for each setup. The computation of the condition number was run in parallel on GPU and took around 8 hours for each pruning rate of each network.

## I.2  Effect of width on conditioning and convergence speed

For a one-hidden layer feed-forward network with ReLU activation, we empirically examined the effect of the hidden width on both the conditioning and the convergence speed when trained on a subset of MNIST ($n = 1000$). The networks were trained with SGD with a fixed learning rate, which was chosen via grid-search, on a single NVIDIA GeForce RTX 3090 GPU. The learning rate was 0.3 for the networks with width 15 and 20 and 0.5 for all the remaining networks. The networks were initialized via the Xavier normal initialization in PyTorch [Paszke et al., 2019]. The training time took less than one hour for all networks. The computation of the condition number was performed in parallel for each width and took around 4 hours in the largest setting. Figure 18 shows the trend that a larger width improves both the conditioning and the convergence speed. Additionally, the condition number is also more stable throughout training for larger widths.

## I.3  Additional experiments on the effect of residual connections

As we have seen previously in Figure 3, the upper bound as a linear combination in Lemma 2 is crucial to get a tight bound on the condition number. Additionally, as illustrated in Figure 19,

---

[3]The implementation of the ViT was based on Code taken from https://github.com/tintn/vision-transformer-from-scratch/tree/main, which was published under the MIT license.

[4]The implementation of the ResNet20, ResNet32 and VGG5 was based on Code taken from https://github.com/jerett/PyTorch-CIFAR10. The author granted explicit permission to use the code.

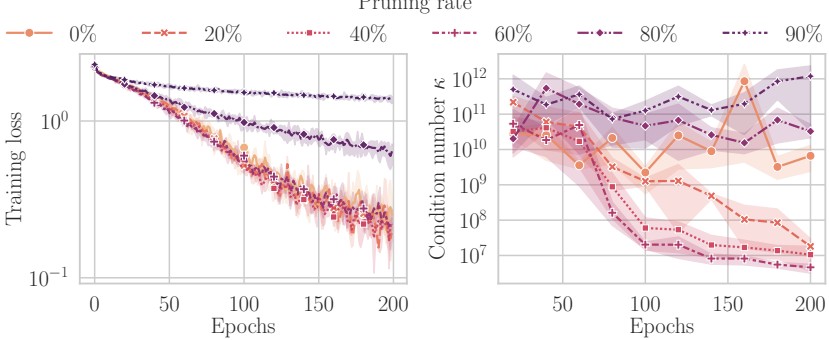

Figure 13: Training loss (left) and condition number (right) for different amounts of pruning at initialization for a Vision Transformer for three seeds. The shaded area in the figures corresponds to one standard deviation from the mean.

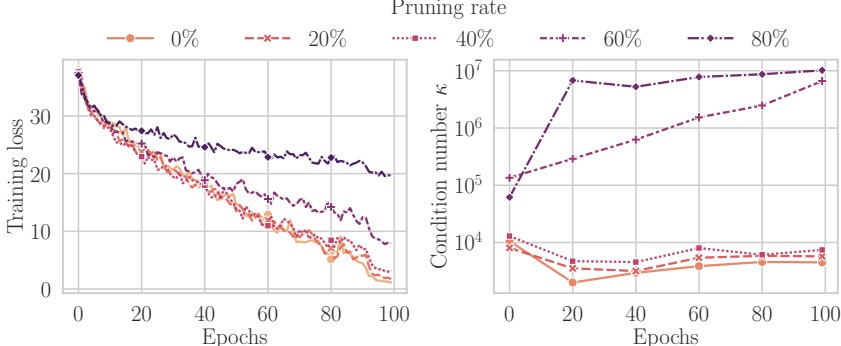

Figure 14: Training loss (left) and condition number (right) for different amounts of pruning at initialization for a ResNet20.

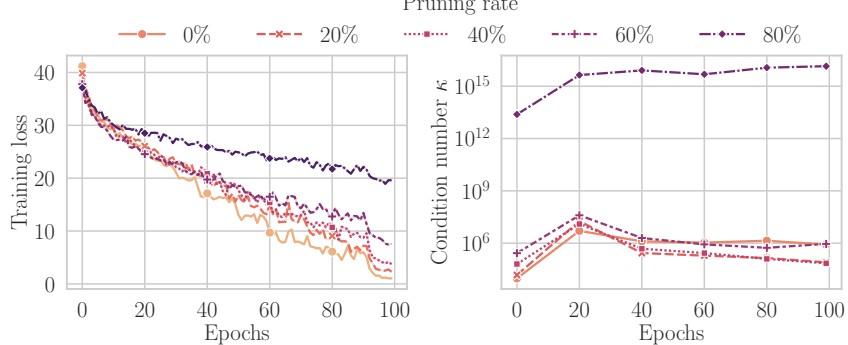

Figure 15: Training loss (left) and condition number (right) for different amounts of pruning at initialization for a ResNet32.

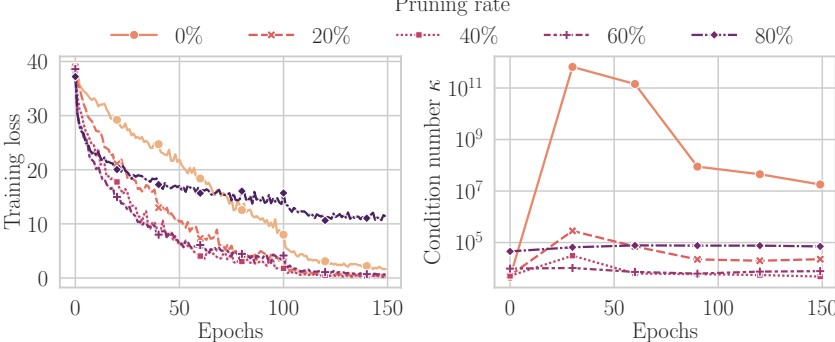

Figure 16: Training loss (left) and condition number (right) for different amounts of pruning at initialization for a VGG5.

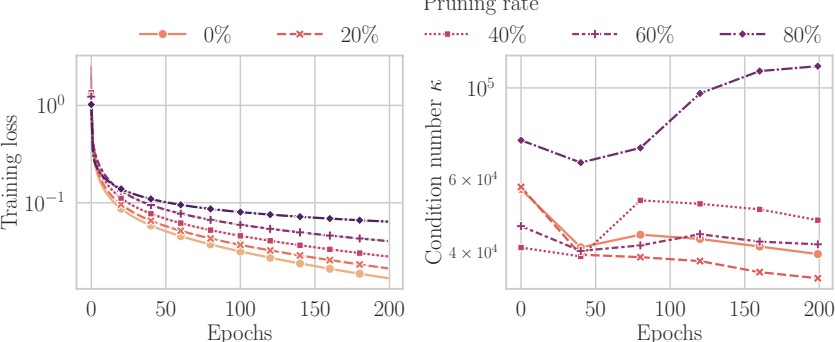

Figure 17: Training loss (left) and condition number (right) for different amounts of pruning at initialization for a Feed-forward network.

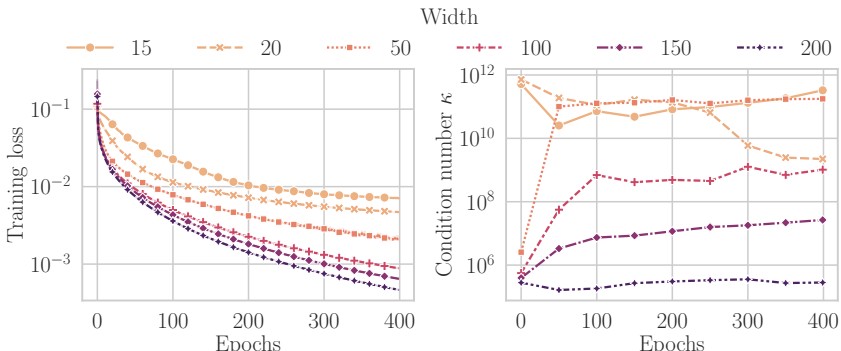

Figure 18: Training loss (left) and condition number (right) for different widths of a one-hidden layer Feed-forward network trained on subset of MNIST ($n = 1000$).

whitening the data is essential to improve the conditioning. Therefore, if not otherwise specified, MNIST LeCun et al. [1998] and Cifar-10 Krizhevsky et al. [2009] will refer to whitened data and we refer to Lemma 2 or its equivalent in Eq. (7) for residual networks as the "upper bound". The shaded area in the figures corresponds to one standard deviation from the mean. If not other specified via the Kaiming normal initialization in PyTorch was used.

As one can see from (8), the condition number for each term in the sum of the first upper bound in (8) improves when $\beta$ increases. This can be seen by the fact that the ratio will be dominated by $\beta$ and will go to 1 for $\beta \to \infty$. This is also what we observe empirically in Figure 20 and Figure 22, where the condition number is smaller for $\beta = 1$ compared to the other two settings, where $\beta = \frac{1}{L} < \frac{1}{\sqrt{L}} < 1$ for deeper networks with $L > 1$.

In Figure 23 the condition number of the GN matrix for a one-hidden layer network with Leaky-ReLU activation is plotted for different values of $\alpha$, where $\alpha = 0$ corresponds to ReLU and $\alpha = 1$ corresponds to a linear activation. We can see how ReLU activation improves the conditioning in the one-hidden layer setting.

The condition number was computed on CPU via the explicit formulas given in Lemma 2 and Eq. (7) and took approximately a few hours per experiment.

## I.4  Further details on experiments compute resources

The total compute for all experiments conducted amounts to approximately less than 100 hours. The main bottleneck is the computation of the condition number, which can be potentially further sped up by making full use of parallel computations on all available GPUs. The full research project required more compute than the experiments reported in the paper due to preliminary experiments.

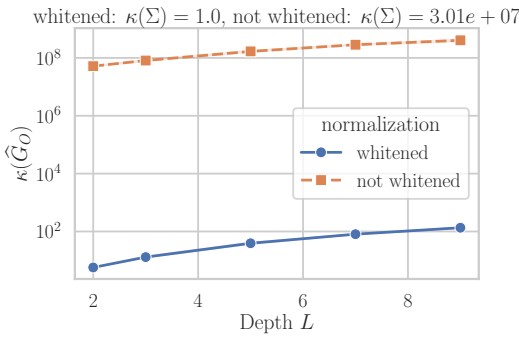

Figure 19: Comparison of the condition number for a Linear Network with hidden width $m = 3100$ with and without whitening Cifar-10 at initialization over three runs. Note, that the $y$-axis is displayed in log scale.

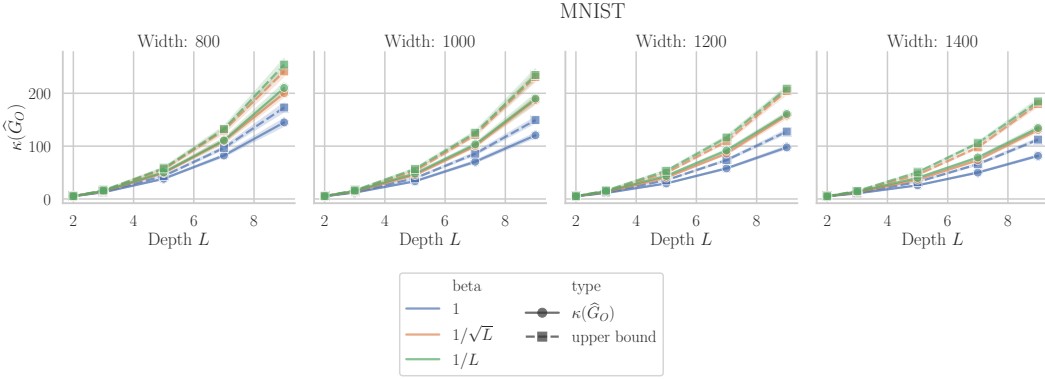

Figure 20: Condition number of outer-product Hessian $\kappa(\widehat{\mathbf{G}}_O)$ and upper bound of a Residual Network for different values of $\beta$ as a function of depth for whitened MNIST at initialization over 20 runs.

Nevertheless, the total compute of the full research project amounts to a run time of similar order of magnitude.

### I.4.1 Computational complexity of computing condition number of Gauss-Newton matrix

In order to compute the condition number we need to compute the eigenspectrum of the GN matrix $\mathbf{G}_O$, which has dimension $p \times p$, where $p$ is the number of parameters or of the matrix $\hat{\mathbf{G}}_O$, which has dimensions $kd \times kd$, where $d$ and $k$ are the input, respectively output dimension of the network. The time complexity of calculating the eigenvalue decomposition has a cubic complexity. That is, in order to compute the condition number, we have a computational complexity of $\mathcal{O}(\min(p, kd)^3)$.

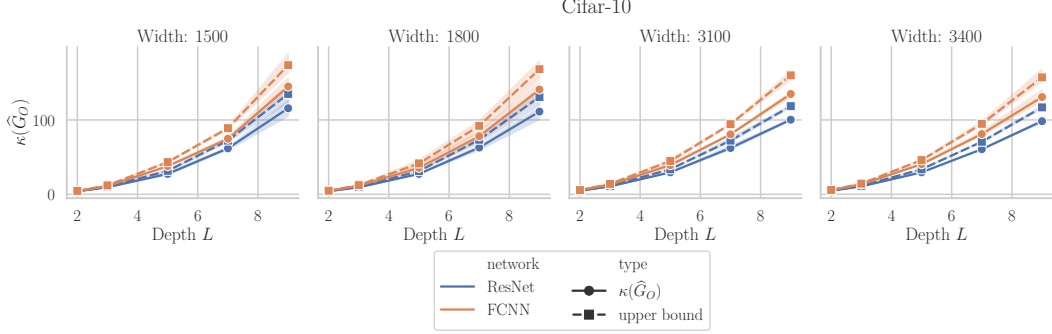

Figure 21: Comparison of condition number of outer-product Hessian $\kappa(\widehat{\mathbf{G}}_O)$ between Linear Network and Residual Network for whitened Cifar-10 at initialization over three runs.

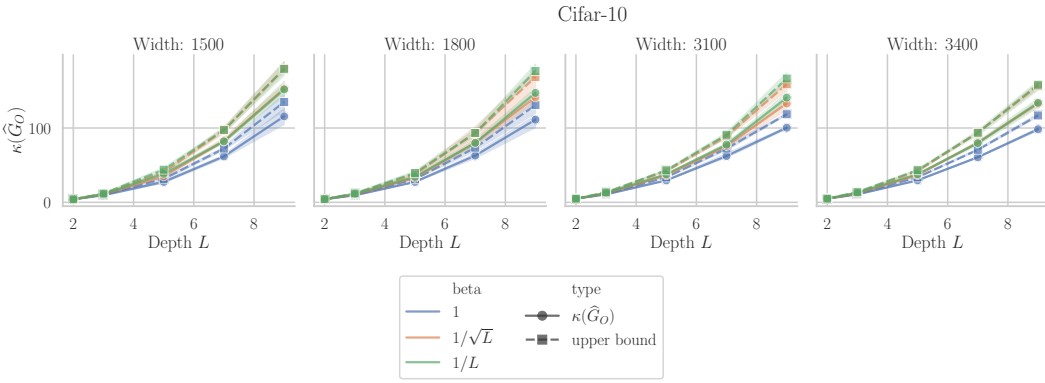

Figure 22: Condition number of outer-product Hessian $\kappa(\widehat{\mathbf{G}}_O)$ and upper bounds for Residual Network with different values of $\beta$ at initialization over three runs for Cifar-10.

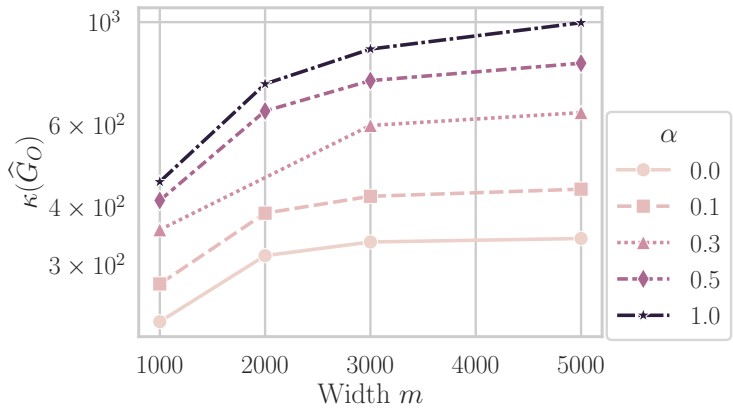

Figure 23: Condition number of a one-hidden layer Leaky-ReLU network for different values of $\alpha$ for whitened MNIST over $n = 500$ data points, where $\alpha = 0$ corresponds to ReLU and $\alpha = 1$ corresponds to a linear activation. Note that the y-axis is log-scaled.

