# OpenReview forum: "Theoretical Characterisation of the Gauss Newton Conditioning in Neural Networks"
_NeurIPS.cc/2024/Conference — NeurIPS 2024 poster_

### Official Review · Reviewer_ewWf · 2024-06-27

**Soundness:** 4
**Presentation:** 3
**Contribution:** 3
**Rating:** 8
**Confidence:** 2

**Summary:**

The authors derived upper bounds for the condition number of the outer product of the jacobian of the neural network output (the Gauss Newton matrix) in the case of deep linear networks, and non-linear networks with a single hidden layer with piecewise-linear activation functions. They empirically evaluate these bounds using MNIST (with the exception of one plot in the main text that used CIFAR-10) and find their bounds reflect the general trend of the empirical condition number as depth (or in some cases width) are varied. The work is motivated by highlighting the importance of Hessian information in the optimisation of neural networks.

**Strengths:**

- The empirical evaluations show that the bounds are actually quite informative, following the general behaviour of the condition number and in some cases being quite tight as well. They point out that if they replace the convex combination in their bound(s) with a maximum, the bound becomes much looser and does not show any behaviour following trends.
- The understanding of neural network loss landscapes and how best to optimise them is still in its relative infancy, so works like this can be valuable.
- Extensive proofs and extra plots are provided in the appendices

**Weaknesses:**

- ~~The paper generally uses the same dataset (MNIST) for almost all the plots, so it's hard to tell if these good behaviour of their bounds hold on other, potentially harder to optimise, problems such as CIFAR-10.~~ **It was pointed out in the rebuttal that CIFAR-10 was included in the appendix.**
- The precise motivation of studying the Gauss Newton is slightly weak. It's not clear to me that these results actually tell us anything substantial about optimising the neural network.
- The authors were not able to extend their bound to the deep non-linear network case.

**Questions:**

- To be clear, on line 82 where we refer to "outer gradient product of the loss function", this is actually the outer gradient product of the network function and has no dependence on the loss function, right?
- Suggestion: In the equation below line 182 (which should probably be numbered) some extra brackets and the use of \cdots rather than \dots would make this expression clearer to read.

**Limitations:**

The authors briefly discuss some future directions their work could be taken in, but do not seriously critique the work they present here.

---

> ### Author Rebuttal · Authors · 2024-08-06
>
> **Weakness 1**:
> > The paper generally uses the same dataset (MNIST) for almost all the plots, so it's hard to tell if these good behaviour of their bounds hold on other, potentially harder to optimise, problems such as CIFAR-10.
>
> **Answer**:
> Thank you for this comment. Please note that we do have experiments using Cifar-10 in the Appendix (see for instance Figure 20 and 21), which showcase that our bounds are also tight in this case. Does this address your comment?
>
> **Weakness 2**:
> The precise motivation of studying the Gauss Newton is slightly weak. It's not clear to me that these results actually tell us anything substantial about optimising the neural network.
>
> **Answer**:
> Thank you for this comment. We would like to answer your comment in two parts:
> - **Preconditioning methods based on GN matrix are successful.** Although the Gauss-Newton matrix $\mathbf{G}_O$ is only an approximation to the full Hessian matrix, it does seem to capture the curvature of the loss very well given the success of many second-order optimization methods based on approximations of the Gauss-Newton matrix, such as K-FAC [Martens and Grosse, 2020], Shampoo [Gupta et al., 2018] or Sophia [Liu et al., 2023].
> - **GN matrix performs better than Hessian as preconditioner in Sophia algorithm.** Particularly interesting is the last method, in which the authors observe that their optimizer based on the Gauss-Newton matrix performs even better than their optimizer based on the full Hessian matrix, implying that the Gauss-Newton matrix is a good preconditioner and captures the curvature of the loss landscape well.
> - As the condition number characterizes the convergence rates of gradient-based methods at least locally, we would like to argue that our results do help in understanding the optimization process of neural networks with gradient-based methods better.
> <!-- Furthermore, we would like to highlight the close connection of the GN matrix to the Neural Tangent Kernel (NTK) and the Jacobian. Therefore, we believe that a better understanding of the GN matrix can also help to provide valuable insights into related objects.  -->
>
> **Weakness 3**
> The authors were not able to extend their bound to the deep non-linear network case.
>
> **Answer**:
> We are indeed not yet able to extend our bounds to deep non-linear networks, which is something that we also mention in the limitation section. Yet we believe that our theoretical bounds on deep linear and residual networks already provide valuable insights on how different choices in the network architecture can affect the condition number of the Gauss-Newton matrix.
>
> **Question 1**:
> To be clear, on line 82 where we refer to "outer gradient product of the loss function", this is actually the outer gradient product of the network function and has no dependence on the loss function, right?
>
> **Answer**:
> Thank you for this question. Yes, you are right, the Gauss-Newton matrix is defined in Eq.(1) (same as defined in prior work discussed above) and it has no dependence on the loss function. As elaborated in lines 84-90, this is precisely the outer product Hessian $\mathbf{H}_O$ when the loss function is the MSE loss, which is what we are considering for most part of the work.
>
> **Question 2**:
> Suggestion: In the equation below line 182 (which should probably be numbered) some extra brackets and the use of $\cdots$ rather than $\dots$ would make this expression clearer to read.
>
> **Answer**:
> Thank you for this suggestion! We will improve the display of the equation to make it more readable.
>
> We kindly request the reviewer to let us know if there are any remaining
> questions they may have. If they find that their queries have been sufficiently addressed, we would greatly appreciate it if they could reconsider their evaluation of our paper.

---

> > ### Comment · Reviewer_ewWf · 2024-08-09
> >
> > Thank you for your comprehensive response.
> >
> > Yes, I see now that you have also run the main experiments on CIFAR-10, so this does address my concern. I wonder whether you could fit both MNIST and CIFAR-10 in figure 5, perhaps side by side? I believe the camera-ready version allows an extra page (please double check) so do consider this in case of acceptance.
> >
> > I am convinced by your explanation of the motivation. On second inspection, it does appear that this is alluded to in the introduction, but perhaps some more explicit references to work like KFAC and Sophia might make the motivation clearer, as you said in your rebuttal.
> >
> > I also agree that it is fair to leave the deep non-linear case to future work.
> >
> > Thank you for clarifying my points of confusion. Please make modifications to wording if you believe it will help readability.
> >
> > Overall, I think my initial score of 6 (weak accept) was slightly harsh. After realising that the experiments were also run on CIFAR-10, I am happy to raise my score to an 8 (strong accept) as I believe this is good work that has been carried out to an excellent standard.

---

### Official Review · Reviewer_6bdM · 2024-06-28

**Soundness:** 2
**Presentation:** 3
**Contribution:** 3
**Rating:** 6
**Confidence:** 4

**Summary:**

This paper examines the condition number of the Gauss-Newton matrix [1] in neural networks. It shows that normalization techniques, such as Batch Normalization [2], initial normalization, skip connections [3], and appropriate layer dimensions, reduce the condition number and therefore enhance the training stability. The objective is to provide insights into the training of deep neural networks by characterizing a new, tight upper bound for the condition number of the Gauss-Newton matrix. The paper primarily focuses on linear neural networks and shallow non-linear neural networks.

[1] SCHRAUDOLPH, Nicol N. Fast curvature matrix-vector products for second-order gradient descent. Neural computation, 2002, vol. 14, no 7, p. 1723-1738.
[2] IOFFE, Sergey et SZEGEDY, Christian. Batch normalization: Accelerating deep network training by reducing internal covariate shift. In : International conference on machine learning. pmlr, 2015. p. 448-456.
[3] HE, Kaiming, ZHANG, Xiangyu, REN, Shaoqing, et al. Deep residual learning for image recognition. In : Proceedings of the IEEE conference on computer vision and pattern recognition. 2016. p. 770-778.

**Strengths:**

This paper provides new theoretical insight into the training dynamics of Deep Neural Networks (DNNs) by examining the conditioning number of the Gauss-Newton Matrix. The experimental results are robust and shed light on often obscure aspects of DNN optimization, such as the impacts of normalization, layer size, residual connections, and ReLU activation function. Meanwhile, the way the authors interpret the experimental results seems reasonable. Importantly, the paper also clearly outlines its limitations in the conclusion.

**Weaknesses:**

The paper is difficult to read and follow due to several typos and unclear ideas. The main contribution is not significant and heavily relies on Singh's work [4]. The authors discuss the K-FAC method [5], which uses the EFIM and not the Hessian, even though it can be an approximation in certain cases where the model's likelihood is in the exponential family [6]. The paper only addresses the Gauss-Newton matrix and not the Generalized Gauss-Newton matrix [6], which could be more relevant, especially when using a cross-entropy loss function instead of MSE. Additionally, the references are outdated (e.g., line 49 and most of the related work), and there are typographical errors, such as "generalized" being repeated on line 140 and "collapse" being misspelled on line 124. There is also a lack of references related to the Gauss-Newton matrix, Kronecker properties PyTorch library, etc...

[4] SINGH, Sidak Pal, BACHMANN, Gregor, et HOFMANN, Thomas. Analytic insights into structure and rank of neural network hessian maps. Advances in Neural Information Processing Systems, 2021, vol. 34, p. 23914-23927.
[5] MARTENS, James et GROSSE, Roger. Optimizing neural networks with kronecker-factored approximate curvature. In : International conference on machine learning. PMLR, 2015. p. 2408-2417.
[6] MARTENS, James. New insights and perspectives on the natural gradient method. Journal of Machine Learning Research, 2020, vol. 21, no 146, p. 1-76.

**Questions:**

1. Could you clarify the statement on line 31: “imagine an entire set of neurons being dead”? What implications does this scenario have for your study?
2. How good is the approximation of the Gauss Newton matrix $\mathbf{G}_O$ without considering the term $\mathbf{H}_F$?
3. Can you also conduct experiments on more challenging datasets like ImageNet [7] or TinyImageNet [8]?
4. Can you develop more about the time and space complexity of the computation of the condition number?
5. Can you explain and rephrase the lines 719 and 720 with the associated Figures 19 and 21?
6. Why did you compute the condition number on CPU rather than on GPUs (line 725)?
7. Can you also explore the relation between te condition number and the batch size?

[7] DENG, Jia, DONG, Wei, SOCHER, Richard, et al. Imagenet: A large-scale hierarchical image database. In : 2009 IEEE conference on computer vision and pattern recognition. Ieee, 2009. p. 248-255.
[8] Le, Ya and Xuan S. Yang. “Tiny ImageNet Visual Recognition Challenge.” (2015).

**Limitations:**

The limitations of the work are well addressed by the authors. I do not believe this work has any particular negative social impact.

---

> ### Author Rebuttal · Authors · 2024-08-07
>
> **Weakness 1**:
> > The paper is difficult to read and follow due to several typos and unclear ideas. The main contribution is not significant and heavily relies on Singh's work [4].
>
> **Answer**:
> We understand the concern of the reviewer regarding the potential overlap with Singh et al. [2021].
> - We would like to clarify that while our work builds on Singh et al. [2021], our **main contribution** is the introduction of **tight upper bounds for the condition number of the Gauss-Newton (GN) matrix** for linear and residual networks of arbitrary depth and width. To the best of our knowledge, this has **not been addressed before in the literature**.
> - More specifically, Singh et al. [2021] derived expressions for the Hessian which we use in our analysis. Therefore, there is an **entirely different focus and a thematic difference** between our current work and the previous work in Singh et al. [2021]. However, obtaining bounds on the GN matrix based on these expressions is not an easy task. We demonstrated that naive bounds are vacuous, while we presented experimental evidence demonstrating that the theoretical bounds are predictive in practice. We would be grateful if the reviewer could elaborate on what ideas are not clear in the paper, we would be happy to provide further details.
>
> **Weakness 2**:
> > The paper only addresses the Gauss-Newton matrix and not the Generalized Gauss-Newton matrix [6], which could be more relevant, especially when using a cross-entropy loss function instead of MSE.
>
> **Answer**:
> Thank you for this insight regarding the Generalized Gauss-Newton matrix.
> - Our current work focuses on the Gauss-Newton matrix to build a solid foundational understanding of its properties, which we believe is important before extending our analysis to more complex scenarios.
> - While we recognize the relevance of the Generalized Gauss-Newton matrix, particularly for cross-entropy loss, addressing this is currently beyond the scope of our work. We believe that this is a key direction for future research and will add this to the discussion on limitations and future work in the final version of the paper.
> - Despite this, our **findings on the Gauss-Newton matrix already provide important insights** into the *initialization and training dynamics of neural networks using MSE loss*, which are valuable in their own right.
>
> **Weakness 3**
> > Additionally, the references are outdated (e.g., line 49 and most of the related work). There is also a lack of references related to the Gauss-Newton matrix, Kronecker properties PyTorch library, etc
>
> **Answer**:
> Thank you for this comment. We apologize for the outdated references. We have now updated our references to cover more recent and additional relevant works, particularly on the Gauss-Newton matrix, normalization techniques, and Kronecker properties. This also includes replacing the outdated references with new ones.
>
> **Weakness 4**:
> > [...] and there are typographical errors, such as "generalized" being repeated on line 140 and "collapse" being misspelled on line 124.
>
> **Answer**:
> We apologize for the issues regarding the readability and typographical errors in the initial submission.
> - We would like to point out however that it is in fact not a typographical error, as Liao and Mahoney [2021] indeed analyze a generalization of the family of generalized linear models, which they called generalized generalized linear models (G-GLM).
> - We have also thoroughly revised our work to correct all typos and improve the clarity and have also rephrased sections, such as lines 140, for better comprehension.
>
> **Question 1**:
> > Could you clarify the statement on line 31: “imagine an entire set of neurons being dead”? What implications does this scenario have for your study?
>
> **Answer**:
> The statement on line 31 refers to the phenomenon where neurons become inactive (that is, produce zero output) due to poor initializations or training dynamics. This will have a direct effect on the eigenspectrum of the GN matrix, which we will illustrate on a simple example.
> Consider a 2-layer linear network $F_{\theta}(\mathbf{x}) = \mathbf{W} \mathbf{V} \mathbf{x}$, where one neuron in the hidden layer is dead, that is output only zeros. Then this is equivalent to the row corresponding to the neuron of the matrix $\mathbf{W}$ to be a zero vector. This directly implies that the rank of $\mathbf{W}$ is reduced by one or that there is another zero eigenvalue in the eigenspectrum. This in turn increases the value of the pseudo condition number $\kappa(\mathbf{W})$ that appears in the upper bound in Eq. (4).
>
> **Question 2**:
> > How good is the approximation of the Gauss Newton matrix 𝐺𝑂 without considering the term 𝐻𝐹?
>
> **Answer**:
> Thank you for this question.
> - The difference between the Gauss Newton matrix $\mathbf{G}_O$ and the Hessian of the loss function $\mathbf{H}_L$ depends on both the residual and the curvature of the network $F\_{\boldsymbol{\theta}}(\mathbf{x})$. Thus, close to convergence when the residual becomes small, the contribution of $\mathbf{H}_F$ will also be negligible and $\mathbf{G}_O$ is essentially equal to $\mathbf{H}_L$.
> - Furthermore, Lee et al. [2019] show that sufficiently wide neural networks of arbitrary depth behave like linear models during training with gradient descent. This implies that the Gauss-newton matrix is a close approximation of the full Hessian in this regime throughout training.

---

> ### Author Response · Authors · 2024-08-07
> **Answers to Questions 3-7**
>
> **Question 3**:
> > Can you develop more about the time and space complexity of the computation of the condition number?
>
> **Answer**:
> Thank you for this question. In order to compute the condition number we need to compute the eigenspectrum of the GN matrix $\mathbf{G}_O$, which has dimension $p \times p$, where $p$ is the number of parameters or of the matrix $\hat{\mathbf{G}}_O$, which has dimensions $kd \times kd$, where $d$ and $k$ are the input, respectively output dimension of the network. The time complexity of calculating the eigenvalue decomposition has a cubic complexity. That is, in order to compute the condition number, we have a computational complexity of $\mathcal{O}(\min(p, kd)^3)$.
>
> **Question 4**:
> > Can you also conduct experiments on more challenging datasets like ImageNet [7] or TinyImageNet [8]?
>
> **Answer**:
> Thank you for this suggestion.
> - **Only marginal gain in insight is expected from experiments on ImageNet.** We would like to argue that we expect only a marginal gain in insights from additional experiments on ImageNet compared to experiments on Cifar-10, which we have already conducted.
> In particular, note that the complexity of a given dataset appears only through the input covariance matrix $\boldsymbol{\Sigma}$, which is a separate factor in the upper bound. Furthermore, as we have mentioned in Remark R1, the effect of the conditioning of the input data on the conditioning of the GN spectra can be largely reduced by whitening or normalizing the input data.
> Thus, the effect that different datasets have on the condition number of the GN matrix can be largely alleviated through preprocessing, which is common practice.
> - **Scaling up experiments is expensive and challenging.** An additional aspect that we want to emphasize is the additional effort to scale up the computation of the condition number, which is challenging and not straightforward, given the time and space complexity elaborated in the previous question.
>
> Given the above explanations, we believe that experiments on ImageNet are less relevant to our work and currently also out of scope.
>
> **Question 5**:
> > Can you explain and rephrase the lines 719 and 720 with the associated Figures 19 and 21?
>
> **Answer**:
> Thank you for this question. As one can see from Eq. (6), the condition number for each term in the sum of the first upper bound in Eq.(6) improves when $\beta$ increases. This can be seen by the fact that the ratio will be dominated by $\beta$ and will go to 1 for $\beta \to \infty$. This is also what we observe empirically in Figure 19 an 21, where the condition number is smaller for $\beta = 1$ compared to the other two settings, where $\beta = 1/L < 1/\sqrt{L} < 1$ for deeper networks with $L > 1$.
>
> **Question 6**:
> > Why did you compute the condition number on CPU rather than on GPUs (line 725)?
>
> **Answer**:
> Thank you for this question.
> - In the case of linear and residual networks (with no activation function), where the GN matrix can be expressed analytically through Eq. (2), we did not find a significant time advantage of running the code on GPU compared to CPU.
> - In the other case, where the GN matrix had to be computed numerically through automatic differentiation (for instance the experiments on pruning weights at initialization), we ran into memory problems with the GPUs. The way we resolved this was to build up the GN matrix through backpropagation on the GPU and move it to CPU to finally compute the condition number. Despite the slow down by moving the GN matrix between GPU and CPU, this still led to a speed up in time.
>
> **Question 7**:
> > Can you also explore the relation between the condition number and the batch size?
>
> **Answer**:
> Thank you for this interesting question!
>
> - Our work does not explicitly consider training dynamics, so generally speaking, our bounds consider $n$ to be the total number of sample points. However, one could also interpret $n$ as the batch size during training, in which case the condition number could be interpreted as the condition number of the "local loss landscape" of a single mini-batch. Note that $n$ only appears implicitly in our theoretical upper bounds through the condition number of the empirical input covariance matrix $\boldsymbol{\Sigma}$.
>
> We kindly request the reviewer to let us know if there are any remaining
> questions they may have. If they find that their queries have been sufficiently addressed, we would greatly appreciate it if they could reconsider their evaluation of our paper.

---

> > ### Comment · Reviewer_6bdM · 2024-08-11
> > **Official Comment by Reviewer 6bdM**
> >
> > Thank you for the authors' comprehensive rebuttal and the additional analyses provided in response to the feedback from myself and the other reviewers.
> >
> > The authors have well addressed my concerns, which has significantly improved my understanding of the paper. Based on this, I will raise my score to 6 (Weak Accept).

---

### Official Review · Reviewer_oXX9 · 2024-07-11

**Soundness:** 3
**Presentation:** 3
**Contribution:** 2
**Rating:** 5
**Confidence:** 4

**Summary:**

This paper characterizes the conditioning of Gauss-Newton (GN) matrix. The contribution of this paper is clear and straightforward: for deep linear networks, it establishes a bound on the condition number of GN matrix, which is further extended to 2-layer ReLU networks. These bounds could be useful in certain scenarios. Numerical experiments are conducted to support the theoretical claims.

**Strengths:**

1. This paper is clearly written and well-organized. The motivation of conducting the proposed research is clearly demonstrated in the introduction, i.e., why it is interesting to study the condition number of GN matrix for certain types of simplified deep neural networks. The main results and empirical results are also clearly presented.
2. Numerical experiments verify that the bounds are tight under conditions imposed by the authors.

**Weaknesses:**

1. My major concern is about the implications of the derived bounds. Specifically, I note that training/learning such as gradient descent learning dynamics, which is crucial in practice, is not involved in deriving the bound. Thus it is hard to see the implication of these derived bounds since they are the same for both before and after training of the model parameters, i.e., it provides no information about the effects of training and fails to characterize the properties of solutions of deep learning models in a specific task. This might even lead the bounds to be meaningless in certain scenarios. For example, in Lemma 1, let $k = 1$, then according to [1],  after enough iterations of gradient descent, the largest singular value $\sigma_{\max} \to \infty$ (as there is no formal definition of $\sigma_{\max}$ in Lemma 1 I assume that my understanding of its definition is correct) while $\sigma_{\min} \to 0$ for both $W$ and $V$, thus the bound blows up and becomes meaningless.

2. I find the introduction part of this paper a bit verbose, e.g., it spends about 4 pages (almost half of the main body) to discuss existing results and related works before presenting details of main results (starting from Lemma 1). I think it would be better to emphasize more about the technical contributions of the current work, which is not clear to me since it seems that many important steps have already been solved by previous works, e.g., Eq. (2), (3), and Lemma 3.

**Reference**

[1]  Ji and Telgarsky. Gradient descent aligns the layers of deep linear networks.

**Questions:**

Could the authors give some implications of the proposed bounds and how we can better use it in practice?

**Limitations:**

The authors addressed their limitations in Section 7. In addition, in my view, the bounds fail to capture the distinctions between the solutions of deep learning models and random parameters such as the initialization, which limits its significance.

---

> ### Author Rebuttal · Authors · 2024-08-06
>
> **Weakness 1**:
> > My major concern is about the implications of the derived bounds. Specifically, I note that training/learning such as gradient descent learning dynamics [...] is not involved in deriving the bound. Thus it is hard to see the implication of these derived bounds since they are the same for both before and after training of the model parameters [...]. This might even lead the bounds to be meaningless in certain scenarios. For example, in Lemma 1, let 𝑘=1, then according to [1], after enough iterations of gradient descent, the largest singular value 𝜎max→∞ [...] while 𝜎min→0 for both 𝑊 and 𝑉, thus the bound blows up and becomes meaningless.
> [1] Ji and Telgarsky. Gradient descent aligns the layers of deep linear networks.
>
> **Answer**:
> Thank you for raising your concerns about the implications of the derived bounds, which we would like to answer in two parts:
>
> 1. **Upper bound remains tight throughout training.** As you correctly pointed out, the bounds are indeed agnostic to the learning dynamics. And it is of course conceivable to find a parametrization of the weight matrices for which the upper bounds become vacuous. However, our empirical results in **Figure 9** in **Appendix D** of the main paper show that the upper bound actually remains tight throughout training, and is thus predictable of the Gauss-Newton condition number throughout training.
> 2. **Reference [1] considers extreme case of linearly separable data.** We have checked reference [1] closely after your comment and would like to note that [1] consider an extreme case, in which they assume that the data is linearly separable, which is of course generally never the case. However, this assumption is essential for the authors in [1] to show the asymptotic weight matrix alignment.
>
> Therefore, we would like to argue that our bounds are indeed useful and relevant in practical settings, as has been illustrated in the point above.
>
> **Weakness 2**:
> > I find the introduction part of this paper a bit verbose, e.g., it spends about 4 pages (almost half of the main body) to discuss existing results and related works before presenting details of main results (starting from Lemma 1). I think it would be better to emphasize more about the technical contributions of the current work, which is not clear to me since it seems that many important steps have already been solved by previous works, e.g., Eq. (2), (3), and Lemma 3.
>
>
> **Answer**:
> Our intention was to provide a comprehensive background and context to ensure that readers with varying levels of familiarity with the topic could fully understand the significance of our contributions. However, your point is well-taken and we will modify the text according to your comment. Thank you for pointing this out.
>
> **Question 1**:
> > Could the authors give some implications of the proposed bounds and how we can better use it in practice?
>
> **Answer**:
> This is a very good question indeed.
> - There are several implications regarding the choice of architecture that are discussed in the paper (for instance the way to scale the width in relation to the depth, and the importance of using normalization layers). The paper also gives a justification of why pruned networks are more difficult to train as they have worse condition number. Although this is not our focus, it is possible that our analysis could inspire better techniques for pruning neural networks.
> - Finally, we also want to mention potential applications in architectural search that is often performed at initialization due to the prohibitive cost of training. [Mellor et al., 2021, Yu et al., 2019, Elsken et al., 2019].
>
> **Limitations 1**:
> > The authors addressed their limitations in Section 7. In addition, in my view, the bounds fail to capture the distinctions between the solutions of deep learning models and random parameters such as the initialization, which limits its significance.
>
> **Answer**:
> Thank you for this comment. We will add this limitation to our discussion in section 7.
>
> We kindly request the reviewer to let us know if there are any remaining
> questions they may have. If they find that their queries have been sufficiently addressed, we would greatly appreciate it if they could reconsider their evaluation of our paper.

---

> > ### Comment · Reviewer_oXX9 · 2024-08-09
> > **Reply to author rebuttal**
> >
> > **Response to rebuttal of weakness 1**
> >
> > My concern still remains. Fig. 9 is not a practical setting: the model for Fig. 9 is a 3-layer linear model and cannot directly perform classification for data that is not linearly separable. The first plot of Fig. 9 only reveals that the loss converges, then
> > 1. if the model fits the data perfectly, then the data is linearly separable, which contradicts the second point of the author response.
> > 2. if the model does not fit the data perfectly, then the first point of the author response is far from sufficient as the training dynamics is actually not a successful one.
> >
> > ---
> > Overall, my point lies in that the proposed methods overlook many aspects of practical settings, therefore the range that the proposed methods can be applied to is rather limited.

---

### Official Review · Reviewer_NTTB · 2024-07-11

**Soundness:** 3
**Presentation:** 3
**Contribution:** 3
**Rating:** 6
**Confidence:** 3

**Summary:**

This paper is dedicated to the theoretical characterization of the condition number of the Gauss-Newton (GN) matrix in neural networks. By studying deep linear networks and two-layer nonlinear networks, the authors establish tight bounds on the GN matrix's condition number and extend this analysis to architectures incorporating residual connections and convolutional layers. The methodology is rigorous, and the experimental validation is thorough, making significant contributions to understanding optimization processes in deep learning.

**Strengths:**

1. This paper deeply studies the properties of Gauss-Newton matrices, especially in terms of condition numbers in deep linear networks and two-layer two-layer nonlinear networks (Leaky ReLU activation), and provides rigorous theoretical derivation and proof.

2. This paper experimentally shows that the width and depth of the network, when the parameters are initialized with a certain distribution, have a strong correlation with the condition number.

**Weaknesses:**

1.The discussion in this paper on the relationship between the Gauss-Newton matrix condition number and the convergence rate of network optimization could be richer.

2.Convergence Rate Analysis in Figure 17: The network's optimal solution and the corresponding minimum loss differ under various settings, making it difficult to analyze the convergence rate from the loss changes. For instance, at epoch 300, the network with a width of 15 may have nearly converged, while the network with a width of 200 still shows a significant downward trend. Thus, it is hard to draw conclusions about the convergence speed.

3.The caption of Figure 9 does not match the subfigures.

4.Figures 12-16 are not referenced or analyzed in the paper.

5.This paper lacks exploration of more general networks and recent network structures.

**Questions:**

1. Can this paper be extended to study the effect of network initialization on convergence rate?

2. Is it more appropriate to adjust the experiment on the effect of condition number and convergence rate in this paper to study different initializations or study the depth of the current network? They are more likely to keep the minimum loss close to each other, and make more use of the analysis of convergence rate.

**Limitations:**

As stated in the weaknesses.

---

> ### Author Rebuttal · Authors · 2024-08-06
>
> **Weakness 1**:
> > Convergence Rate Analysis in Figure 17: The network's optimal solution and the corresponding minimum loss differ under various settings, making it difficult to analyze the convergence rate from the loss changes. [...] Thus, it is hard to draw conclusions about the convergence speed.
>
> **Answer**:
> Thank you for pointing this out.
> - To investigate this further, we have rerun the same experiment for 2000 epochs. The result can be found in Figure 2 of the attached PDF.
> - As the Reviewer pointed out, we indeed observe that the two networks with the smallest width converge to a suboptimal loss. Nevertheless, we *still observe the **connection** between a **smaller condition number** and a **faster convergence rate** for the remaining four networks*.
>
> **Weakness 2**:
> > The discussion in this paper on the relationship between the Gauss-Newton matrix condition number and the convergence rate of network optimization could be richer.
>
> **Answer**:
> Thank you for this comment.
> - As discussed above, we have **rerun one experiment** to investigate the connection between the convergence rate and the Gauss-Newton matrix condition number more closely.
> - Moreover, we discuss below as an *Answer* to your **Question 1** how our work can be extended to study the **effect of the condition number at initialization on the convergence rate**. We hope that these additional discussions have enriched the discussion on the relationship between the condition number of the GN matrix and the convergence rate of network optimization.
>
> **Weakness 3**:
> > The caption of Figure 9 does not match the subfigures.
>
> Thank you for pointing this out. We have corrected the caption now.
>
> **Weakness 4**:
> > Figures 12-16 are not referenced or analyzed in the paper.
>
> Thank you for this comment. We have now added the reference to Figures 12-16 in appendix I.1. All experiments, except the VGG experiment, showcase that there is indeed a connection between the condition number of the network and the convergence speed during training.
>
> **Weakness 5**:
> > This paper lacks exploration of more general networks and recent network structures.
>
> **Answer**:
> It would indeed be interesting to extend our analysis to other architectures, such as CNNs or Transformers.
> - **Straight forward extension to CNN.** We would like to note that our results can already be directly extended to CNNs by making use of the fact that the convolution operation can be reformulated in the form of a matrix-vector product using Toeplitz matrices, in which case we can apply the same analysis as for fully connected networks, which we also mention in a remark in line 266 and further elaborate in Appendix B. Previous work, such as Pinson et al. [2023] or Singh et al. [2022] have also studied properties of linear CNNs, such as its rank, making it an interesting object of study.
> - **Empirical results on CNN.** We addtionally provide empirical results for the condition number of the GN matrix of linear CNNs at initialization in Figure 1 of the attached PDF of the general response, where we examine the effect of kernel size and number of filters on the condition number of the GN matrix.
> - **MLPs in Transformers suggest potential transferability of some theoretical results.** The theoretical analysis of the condition number in Transformers would indeed be very intriguing, although we are currently unable to derive bounds for this setting. We would like to mention that previous work. Nevertheless, we would like to point out that MLPs make up a large part of standard transformer architectures and thus we expect that some theoretical results will also carry over to the Transformer setting.

---

> ### Author Response · Authors · 2024-08-06
> **Answers to Question 1 and 2**
>
> **Question 1**:
> > Can this paper be extended to study the effect of network initialization on convergence rate?
>
> **Answer**:
> This is a very interesting question. As the condition number is a very local property, it is in general hard to connect the conditioning at network initialization to a global convergence rate. However, we would like to argue below that an ill-conditioned network initialization will still affect the rate of convergence for gradient descent (GD) in the initial phase of training. For this we will present a modified analysis of GD for strongly convex functions, where we use local constants $\mu(k)$ and $L(k)$ instead of the global smoothness and Lipschitz constant, respectively.
> Let us denote the Lipschitz constant by $L$ and the smoothness constant by $\mu$. Furthermore, let the step size be such that $\eta_k \leq \frac{1}{L}$. Then by the definition of gradient descent, we have:
>
> \begin{align*}
>  ||\boldsymbol{\theta}\_{k+1} - \boldsymbol{\theta}^*  ||^2 &=  ||\boldsymbol{\theta}_k - \boldsymbol{\theta}^* - \eta_k \nabla f(\boldsymbol{\theta}_k)  ||^2 \\\\
> &=  ||\boldsymbol{\theta}_k - \boldsymbol{\theta}^*  ||^2 -2 \eta_k \nabla f \left( \boldsymbol{\theta}_k \right)^\top \left( \boldsymbol{\theta}_k - \boldsymbol{\theta}^* \right) + \eta_k^2  ||\nabla f \left( \boldsymbol{\theta}_k \right)  ||^2  \\\\
> & \stackrel{\text{Strong convexity}}{\leq} (1-\eta_k \mu)  ||\boldsymbol{\theta}_k - \boldsymbol{\theta}^*  ||^2 - 2\eta_k (f(\boldsymbol{\theta}_k) - f(\boldsymbol{\theta}^*)) + \eta_k^2  ||\nabla f \left( \boldsymbol{\theta}_k \right)  ||^2  \\\\
> & \stackrel{\text{Smoothness}}{\leq} (1-\eta_k \mu)  ||\boldsymbol{\theta}_k - \boldsymbol{\theta}^*  ||^2 - 2\eta_k (f(\boldsymbol{\theta}_k) - f(\boldsymbol{\theta}^*)) + 2 \eta_k^2 L (f(\boldsymbol{\theta}_k) - f(\boldsymbol{\theta}^*)) \\\\
> &= (1-\eta_k \mu)  ||\boldsymbol{\theta}_k - \boldsymbol{\theta}^*  ||^2 - 2 \eta_k (1 - \eta_k L) (f(\boldsymbol{\theta}_k) - f(\boldsymbol{\theta}^*))
> \end{align*}
>
> Since we assumed that $\eta_k \leq \frac{1}{L}$, the last term is negative. Therefore:
>
> \begin{equation}
>  ||\boldsymbol{\theta}_{k+1} - \boldsymbol{\theta}^*  ||^2 \leq (1-\eta_k \mu)  ||\boldsymbol{\theta}_k - \boldsymbol{\theta}^*  ||^2
> \end{equation}
>
>
> So by recursively applying Eq. (1) and replacing $\mu$ by the local smoothness constants $\mu(k)$:
> \begin{equation}
>  ||\boldsymbol{\theta}_k - \boldsymbol{\theta}^*  ||^2 \leq  \prod\_{i=0}^{k-1} (1-\eta_i \mu(i))  ||\boldsymbol{\theta}_0 - \boldsymbol{\theta}^*  ||^2
> \end{equation}
>
> One can clearly see the effect of $\mu(0)$ in the bound, which is even more dominant when $\mu(k)$ changes slowly. Of course, the effect of $\mu(0)$ attenuates over time, and that's why we are talking about a local effect. However, one should keep in mind that overparametrization leads the parameter to stay closer to initialization (at least in the NTK regime). We are happy to add a detailed discussion in the final version of the paper.
>
> **Question 2**:
> > Is it more appropriate to adjust the experiment on the effect of condition number and convergence rate in this paper to study different initializations or study the depth of the current network? They are more likely to keep the minimum loss close to each other, and make more use of the analysis of convergence rate.
>
> **Answer**:
> We are not sure what the Reviewer is asking precisely and kindly ask the Reviewer to rephrase the question, so we can dedicate an answer to it during the Author-Reviewer discussion phase.
>
> We kindly request the reviewer to let us know if there are any remaining questions they may have. If they find that their queries have been sufficiently addressed, we would greatly appreciate it if they could reconsider their evaluationof our paper.

---

> > ### Comment · Reviewer_NTTB · 2024-08-13
> >
> > Thank you for your explanation, which has been generally helpful. Since my concerns have been resolved, I now hold a favorable view of this work and would like to raise my rating to 6.

---

### Author Rebuttal · Authors · 2024-08-06

Dear Reviewers,

we would like to thank you for the time that you have committed to reviewing our work and for the questions and comments, which have helped to enhance our work considerably.

We are pleased to report that we were able to address almost all of your comments and questions (except one question which we hope **Reviewer NTTB** can clarify), which we summarize below:

1. - As suggested by **Reviewer NTTB** we have extended our discussion on the relationship between the Gauss-Newton matrix condition number and the convergence rate of network optimization, which we substantiate with an experiment that we discuss below in more detail.
    - Additionally, we have conducted another experiment on linear CNNs, which highlights the empirical applicability of our bounds in the CNN setting.
2. As requested by **Reviewer oXX9** we have elaborated the practical relevance of our derived bounds, which can be seen in Figure 9 in appendix D of our main paper, and further discussed how these bounds can be better used in practice.
3. As requested by **Reviewer 6bdM** we have highlighted the novelty of our contribution and clarified questions regarding the actual computation of the condition number (e.g. CPU vs. GPU, computational complexity).
4. As requested by **Reviewer ewWf**, we provide further motivation for studying the Gauss-Newton matrix and its relevance for understanding the optimization process of neural networks.
5. We clarified some formulations, corrected some minor typos, and promised to update parts of our references.

We believe that our revision highlights the novelty of our contribution, and that the additional experiments, which we provide, support our claim. We are looking forward to the Author-Reviewer discussion period.
In the meantime, we kindly ask you to re-evaluate our paper and consider raising your scores and confidence in your assessments.

Best regards,

The Authors


### Further experiments
- **Experiments on CNNs.** As requested by **Reviewer NTTB** we conducted additional experiments on the condition number of the Gauss-Newton matrix of linear CNNs at initialization, which can be found in Figure 1 of the attached PDF. We examine the effect of kernel size and number of filters on the condition number of the GN matrix. We observe a trend, where the number of filters increases the condition number (in analogy to depth in MLPs) and the kernel size improves conditioning (in analogy to width in MLPs). This highlights the empirical applicability of our bounds in the CNN setting.
- **Connection between condition number of GN matrix and convergence speed.** As has also been requested by **Reviewer NTTB** we have rerun an experiment, which evaluates the convergence speed of a 2-layer ReLU network with varying width, for more epochs. The result can be found in Figure 2 of the attached PDF. Although the two networks with small widths converge to a suboptimal minimum and should be discarded from the discussion on convergence speed, we can still observe the connection between a smaller condition number and a faster convergence rate for the remaining four networks.

---

### Decision · Program_Chairs · 2024-09-25

**Decision:**

Accept (poster)

**Comment:**

This paper provides a theoretical characterization of the condition number of the Gauss-Newton (GN) matrix in neural networks, establishing tight bounds for linear networks and extending the analysis to some non-linear, residual, and convolutional architectures. This paper provides interesting analysis and will benefit the community, and therefore I recommend acceptance.